# Parallel visual circuitry in a basal chordate

Matthew J Kourakis[1†], Cezar Borba[2†], Angela Zhang[3], Erin Newman-Smith[1,2], Priscilla Salas[2], B Manjunath[1], William C Smith[1,2]*

[1]Neuroscience Research Institute, University of California, Santa Barbara, Santa Barbara, United States; [2]Department of Molecular, Cell and Developmental Biology, University of California, Santa Barbara, Santa Barbara, United States; [3]Department of Electrical and Computer Engineering, University of California, Santa Barbara, Santa Barbara, United States

**Abstract** A common CNS architecture is observed in all chordates, from vertebrates to basal chordates like the ascidian *Ciona*. *Ciona* stands apart among chordates in having a complete larval connectome. Starting with visuomotor circuits predicted by the *Ciona* connectome, we used expression maps of neurotransmitter use with behavioral assays to identify two parallel visuomotor circuits that are responsive to different components of visual stimuli. The first circuit is characterized by glutamatergic photoreceptors and responds to the direction of light. These photoreceptors project to cholinergic motor neurons, via two tiers of cholinergic interneurons. The second circuit responds to changes in ambient light and mediates an escape response. This circuit uses GABAergic photoreceptors which project to GABAergic interneurons, and then to cholinergic interneurons. Our observations on the behavior of larvae either treated with a GABA receptor antagonist or carrying a mutation that eliminates photoreceptors indicate the second circuit is disinhibitory.
DOI: https://doi.org/10.7554/eLife.44753.001

*For correspondence:
w_smith@ucsb.edu

†These authors contributed equally to this work

Competing interests: The authors declare that no competing interests exist.

## Introduction

Ascidians, including members of the widely-studied *Ciona* genus, have a biphasic life cycle. At the start of their life, most ascidians spend their first few days as free-swimming tadpole larvae. It is at the larval stage that ascidians display unmistakable chordate traits, including a prominent notochord running the length of a muscular tail and a dorsal central nervous system (CNS). Ascidian morphology abruptly changes when the larvae attach via their adhesive palps and undergo metamorphosis, the product of which is a sessile filter-feeding juvenile with little resemblance to the larva (*Satoh, 1994*). Largely because of their conserved chordate body plan, ascidian embryos and larvae have been subjects of extensive investigation (*Satoh, 2014*). Ascidian larvae are small in comparison to vertebrate larvae, with the *Ciona* larva measuring only about 1 mm in length, with a total of ~2,600 cells (*Satoh, 2014*). Accordingly, the larval *Ciona* CNS is equally simple, having ~170 neurons (*Ryan et al., 2016*). Despite this simplicity, *Ciona* larvae display a range of integrated behaviors, including negative gravitaxis and phototaxis, and a response to dimming ambient light, all mediated by central sensory neurons. Also well documented is a mechanosensory/touch response, and possibly chemosensation, mediated by peripheral sensory neurons (*Ryan et al., 2018*). Anatomically, the *Ciona* larval CNS is comprised anteriorly of the *brain vesicle* (BV; also known as the *sensory vesicle*), a region homologous to the vertebrate forebrain and midbrain, followed by the *neck* region, a homolog of the vertebrate midbrain/hindbrain junction (*Figure 1*). Immediately posterior to the neck is the *motor ganglion* (MG; also known as the *visceral ganglion*). The MG is thought to be homologous to the vertebrate hindbrain and/or spinal cord, and contains ten motor neurons (MN) as well as a number of interneurons – including the two *descending decussating neurons* (ddN) which have been equated with vertebrate Mauthner cells which mediate the startle response (*Ryan et al.,*

2017). With the completion of a synaptic connectome from one larva, the *Ciona* larval nervous system is now one of the best described, with the connectome providing a detailed and quantitative connectivity matrix of the 6618 chemical and 1206 electrical CNS synapses (*Ryan et al., 2016*).

The *Ciona* visual system is the best characterized of the larval sensory systems (*Kusakabe and Tsuda, 2007*; *Oonuma et al., 2016*). The larval photoreceptors are ciliary, like those of vertebrates, and the photoreceptor opsins and visual cycle systems are, likewise, similar to those found in the vertebrate retina (*Kusakabe and Tsuda, 2007*; *Kusakabe et al., 2001*). The primary photoreceptive organ of *Ciona* is the ocellus, which consists of two groups of photoreceptors, three lens cells, and one pigment cell. The first group of photoreceptors (called here, PR-I) is comprised of 23 cells and clustered around the ocellus pigment cell (*Figure 1*). The opsin-containing outer segments of the PR-Is project into the cup shaped pigment cell, making this group sensitive to the direction of incident light and thereby mediating negative phototaxis (*Horie et al., 2008a*; *Salas et al., 2018*). The second group of ocellus photoreceptors (PR-II, *Figure 1*) is comprised of seven cells and is adjacent and anterior to the PR-Is, and is not associated with the pigment cell. The PR-II cluster mediates the light dimming response, likely with a contribution from the PR-Is, by evoking highly tortuous and leftward-biased swims (*Salas et al., 2018*). There is a third set of six photoreceptors (PR-IIIs) distal to the ocellus of unknown function, although they are not involved in phototaxis or the dimming response (*Horie et al., 2008a*), nor do they make extensive connections to interneurons, as do the PR-I and -IIs (*Ryan et al., 2016*).

The relationship between the ascidian ocellus and visual systems in other chordates is not fully resolved. Vertebrates are characterized by the presence of paired lateral (*i.e.*, retinal) eyes, as well as an unpaired medial/pineal eye (*Lamb et al., 2007*). The cephalochordate *Amphioxus* by contrast, has four distinct photoreceptive organs (*Pergner and Kozmik, 2017*). The amphioxus *frontal eye* has been proposed as homologous to the vertebrate lateral eyes, while the *lamellar body* is thought to be homologous to the vertebrate pineal organ (*Pergner and Kozmik, 2017*; *Vopalensky et al., 2012*). The other two amphioxus photoreceptor types, the *dorsal ocelli* and the *Joseph cells*, are thought, based on a number of criteria, including their rhabdomeric morphology- which differs from the ciliary morphology of vertebrate and ascidian photoreceptors- to be vestiges of a more primitive photoreceptive system. Based on various criteria the ascidian ocellus has been proposed as having homology to either the medial or the retinal eyes (*Kusakabe et al., 2001*; *Lamb et al., 2007*).

The *Ciona* larval connectome predicts the neural circuity linking photoreceptors to motor activation (*Ryan et al., 2016*). *Figure 1* shows the simplified minimal visuomotor circuit in which neurons of the same type are clustered (*e.g.*, photoreceptors) and the number of neurons of each type is indicated in parentheses. The full connectivity for the visuomotor circuit showing all neurons along with chemical and gap junction/electrical synapses (and their relative strengths) is shown in *Figure 1—figure supplements 1* and *2*, respectively (derived from data tables in [*Ryan et al., 2016*]). As shown in *Figure 1—figure supplement 2*, gap junctions are few and relatively small in the BV and become more prominent in the MG.

The minimal circuit shows the PR-I and –II photoreceptors synapsing primarily onto two classes of relay neurons (RNs) in the posterior BV (pBV). The six *photoreceptor RNs* (prRN) receive input exclusively from the PR-I photoreceptors and then project posteriorly to the paired right/left *MG interneurons* (three on each side; MGIN in *Figure 1* and *Figure 1—figure supplement 1*). A second cluster of eight RNs, the *photoreceptor ascending MG RNs* (pr-AMG RN) are postsynaptic to both the PR-I and PR-II photoreceptors, and are so-named because they, unlike the prRNs, receive input from the *ascending MG peripheral interneurons* (AMG neurons; not shown in *Figure 1*). There are also extensive synaptic connections between the pr-AMG RNs and the prRNs. Like the prRNs, the pr-AMG RNs project posteriorly to the left and right MGINs. The MGINs in turn synapse onto the paired right and left motor neurons (five on each side). The *Ciona* connectome thus predicts a complete visuomotor circuit from photoreceptors to muscle target cells, and provides a valuable comparative model to their chordate cousins, the vertebrates and cephalochordates (*Vopalensky et al., 2012*; *Nilsson, 2009*; *Suzuki et al., 2015*), as well as to other well-described but much more distantly related models such as *Drosophila* and *Platynereis* (*Eichler et al., 2017*; *Larderet et al., 2017*; *Randel et al., 2014*).

In the current study, we report that the *Ciona* PR-I and PR-II visual circuits operate by very different logic. We find that the PR-I circuit is an excitatory relay from glutamatergic photoreceptors signaling to AMPA receptors on cholinergic prRNs. On the other hand, the PR-II circuit initiates with

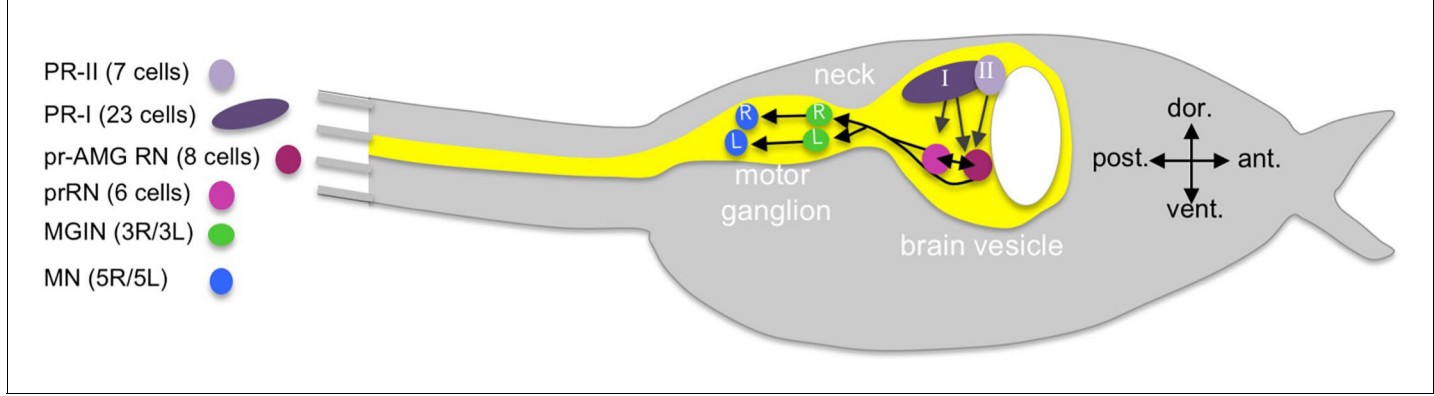

**Figure 1.** Cartoon of a *Ciona* tadpole larva with outline of the central nervous system. The minimal visuomotor circuit is shown with circles representing classes of neurons with the number of cells of each class indicated in the parentheses of the key. Abbreviations: dor., dorsal; vent., ventral; ant., anterior; post., posterior; PR-II, photoreceptor group II; PR-I, photoreceptor group I; pr-AMG RN, photoreceptor ascending motor ganglion relay neuron; prRN, photoreceptor relay neuron; MGIN, motor ganglion interneuron; MN, motor neuron. L, left; R, right. Cell types are color coded according to *Ryan et al. (2016)*.

DOI: https://doi.org/10.7554/eLife.44753.002

The following figure supplements are available for figure 1:

**Figure supplement 1.** Chemical synapse connectivity of minimal visuomotor system of *Ciona*.
DOI: https://doi.org/10.7554/eLife.44753.003

**Figure supplement 2.** Electrical synapse connectivity of minimal visuomotor system of *Ciona*.
DOI: https://doi.org/10.7554/eLife.44753.004

GABAergic photoreceptors synapsing onto GABAergic pr-AMG RNs. Both the sequential array of GABAergic neurons in this pathway and the behavior of larvae treated with GABA inhibitors, or carrying a mutation that misspecifies the anterior BV, support a model in which this circuit is disinhibitory.

## Results

### Glutamatergic and GABAergic photoreceptors

The *Ciona* connectome provides a detailed description of chemical synapse connectivity but it provides no information on neurotransmitter (NT) use. While the expression of genes in the *Ciona* CNS and PNS that mark NT use [*e.g.*, vesicular glutamate transporter (VGLUT), vesicular GABA transporter (VGAT), tyrosine hydroxylase (TH), and vesicular acetylcholine transporter (VACHT) for glutamatergic, GABAergic/glycinergic, dopaminergic and cholinergic neurons, respectively] has been extensively reported (*Brown et al., 2005*; *Horie et al., 2008b*; *Moret et al., 2005*; *Pennati et al., 2007*; *Takamura et al., 2010*), finding exact matches of expression patterns to neurons, or groups of neurons, in the connectome is not always possible. For example, the ocellus is reported to have widespread VGLUT expression, indicating that the *Ciona* photoreceptors, like those of vertebrates are glutamatergic (*Horie et al., 2008b*). However, the expression domains of both VGAT and glutamic acid decarboxylase (GAD) are suggestive of a subpopulation of GABAergic/glycinergic photoreceptors (*Yoshida et al., 2004*; *Zega et al., 2008*), although the identities of these cells within the ocellus is not known. To investigate this further, fertilized eggs from a stable transgenic *Ciona* line expressing *kaede* fluorescent protein under the VGAT promoter (pVGAT > kaede) (*Horie et al., 2011*) were microinjected with a pOpsin1 > red fluorescent protein (RFP) construct (*Kusakabe et al., 2001*; *Kusakabe et al., 2004*) (*Figure 2a*; n = 5 larvae observed). We observed a subset of photoreceptors coexpressing the two fluorescent markers both at the anterior and ventral sides of the ocellus (white and orange arrowheads, respectively). In the field of view shown in *Figure 2a*, the eminens cells is also evident due to its expression of VGAT (white arrow), in agreement with earlier reports of GAD expression (*Takamura et al., 2010*). To investigate BV VGLUT and VGAT expression in greater detail we used Hybridization Chain Reaction in situ (HCR in situ)

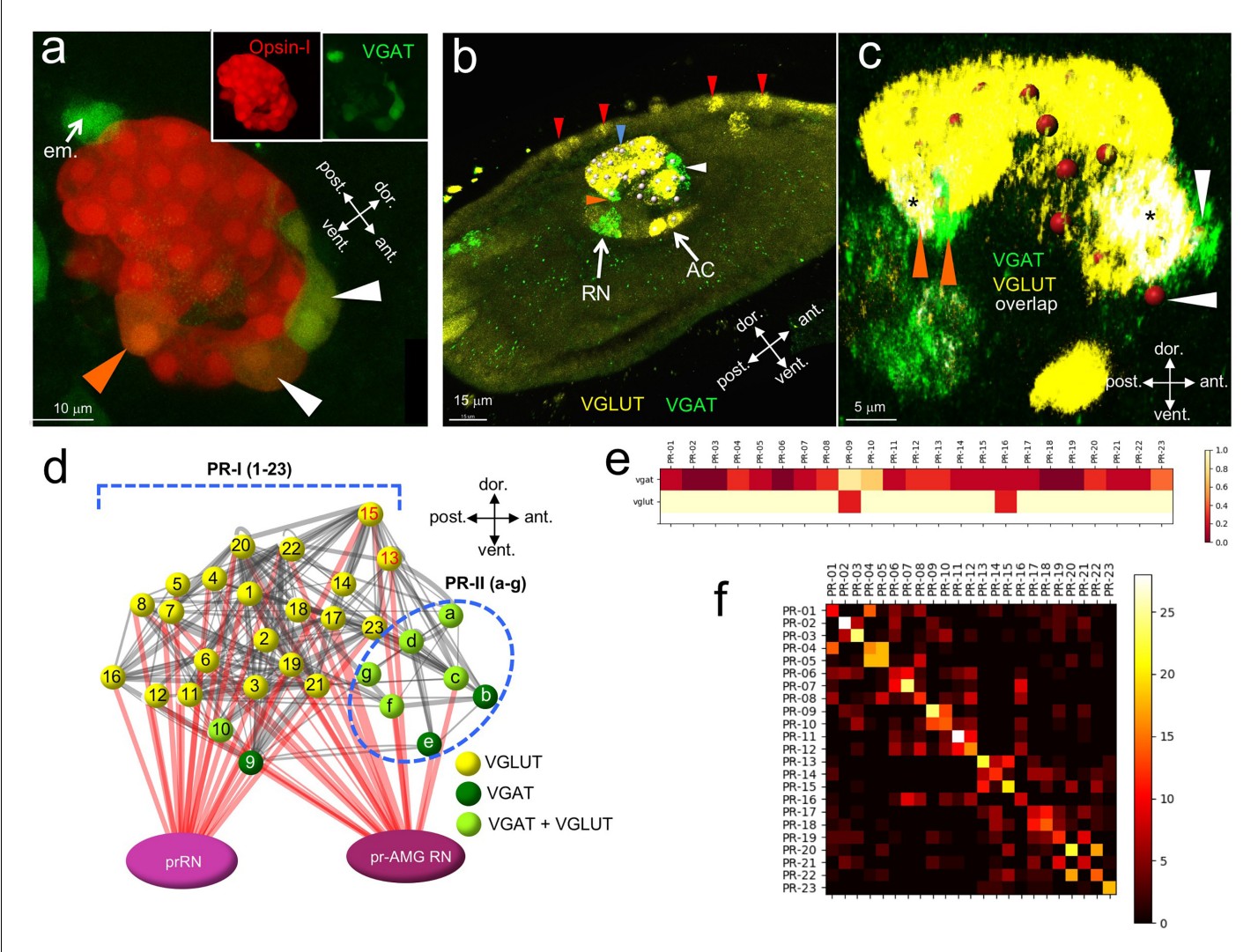

**Figure 2.** Neurotransmitter use in the ocellus. (a) Coexpression of opsin and VGAT reporter constructs in the ocellus (white and orange arrowheads). Insets show expression of Opsin-1 and VGAT individually. (b) Expression of VGLUT and VGAT in the brain vesicle and epidermis by in situ hybridization. VGAT was observed in an anterior (white arrowhead) and posterior (orange arrowhead) domain of the ocellus. Blue arrowhead indicates VGLUT expression in the ocellus, and red arrowheads indicate VGLUT-expressing epidermal sensory neurons. (c) Posterior VGAT-expression in the ocellus consists of two cells (orange arrowheads), one exclusively expressing VGAT, and one coexpressing VGAT and VGLUT. Two cells in the anterior exclusively express VGAT (white arrowheads). Nuclei are shown as red spheres. Asterixis indicate overlap of VGAT and VGLUT. (d) Neurotransmitter predictions color-coded on a schematic diagram of the ocellus photoreceptors. Lines between photoreceptors indicate chemical synaptic connections taken from *Satoh (1994)*, with red lines indicating projections to the relay neurons. (e) Heat map of neurotransmitter predictions from registration for photoreceptor group I (cells 01–23). Scale assigns color to proportion of iterations predicting VGAT or VGLUT within a particular cell. (f) Confusion matrix of registration of photoreceptor group I cells (cells 01–23). High values (light colors) in the diagonal indicate higher confidence. Abbreviations: dor., dorsal; vent., ventral; ant., anterior; post., posterior; em., eminens cell; RN, relay neuron; AC, antenna cells; pr-AMG RN, photoreceptor ascending motor ganglion relay neuron; prRN, photoreceptor relay neuron; VGAT, vesicular GABA transporter; VGLUT, vesicular glutamate transporter; PR-I, photoreceptor group I (01–23).

DOI: https://doi.org/10.7554/eLife.44753.005

The following figure supplement is available for figure 2:

**Figure supplement 1.** Neurons in the visuomotor circuit postsynaptic to the Group-I Photoreceptors (PR1-PR23).
DOI: https://doi.org/10.7554/eLife.44753.006

(*Choi et al., 2018*) (*Figure 2b*). In agreement with previous reports (*Horie et al., 2008b*) we observed VGLUT expression in the ocellus (blue arrowhead), the two otolith antenna cells (AC), and in epidermal sensory neurons (red arrowheads). Consistent with the above transgenic data, VGAT was expressed in two separate clusters within the ocellus (white and orange arrowheads), as well as in a separate group of BV neurons outside the ocellus corresponding to previously described VGAT-positive neurons that project axons to the MG (*Yoshida et al., 2004*) (labeled in *Figure 2b* as RNs, see next section). The anterior VGAT-expressing photoreceptor cluster consisted of 7 cells (±1 cell, n = 17 larvae), while the posterior group consisted of two cells (n = 20 larvae). We also observed a subset of the VGAT-expressing cells in both the anterior and posterior clusters that also expressed VGLUT (*Figure 2c and d*). In the anterior cluster, we observed that the 2 cells (±1; n = 4 VGAT/VGLUT double in situ larvae) at the anterior edge exclusively expressed VGAT (white arrowheads in *Figure 2c*), while the four cells immediately posterior to these cells co-expressed VGAT and VGLUT. In the posterior cluster we observed in all samples (n = 5) that one of two cells co-expressed VGAT and VGLUT, while the other only expressed VGAT (*Figure 2c*, orange arrowheads).

Given the cellular anatomy of the *Ciona* ocellus, with seven PR-IIs anterior and 23 PR-Is posterior (*Ryan et al., 2016*; *Horie et al., 2008a*), and our transgenic and HCR in situ results, we assign the PR-IIs as being VGAT-positive, with a subset co-expressing VGLUT (*Figure 2d*). The anterior/ventral location of the two VGAT-only PR-IIs suggest that they are PR-b and –e (*Figure 2c and d*). By contrast, the majority of the PR-Is are exclusively glutamatergic with the exception of two ventral cells, one co-expressing VGAT and VGLUT and the other expressing only VGAT. While the identities of the PR-II subpopulations were evident from the ocellus anatomy, the identities of the two VGAT-expressing PR-Is were initially less clear. To get a better indication of the identities of these two cells we performed a registration of cell centroids from multiple in situ datasets (n = 11) with the centroids from the connectome serial section electron microscopy (ssEM) dataset. This registration would only be meaningful if there is strong stereotypy in the number and position of the neurons among *Ciona* larvae. The ocellus photoreceptor somata and their outer segments are known to be arranged in rows, suggesting an ordered cellular architecture (*Ryan et al., 2016*; *Horie et al., 2008a*). Moreover, we reasoned that stereotypy in the ocellus, if present, should be evident when registering VGAT- and VGLUT-expressing cells, both across multiple in situ-stained larvae, and individually to the ssEM photoreceptor centroids. Convergence of NT type with registered photoreceptors (both between HCR in situ samples and between these and the ssEM sample) would be taken as evidence of stereotypy, and of the validity of making NT use predictions.

For photoreceptor registration, DAPI-stained ocellus nuclei from VGAT and VGLUT in situ labeled larvae were segmented from 3D image stacks to serve as cell centroids (nuclei are indicated in *Figure 2c* as red spheres). Based on the in situ signal, each centroid was designated as VGAT or VGLUT, or both. Finally, the antenna cell and ddN nuclei in the image stacks were segmented to serve as anchor points for registration. Registration of the segmented HCR in situ nuclei to each other and to the connectome PR-I nuclei was done according to (*Myronenko and Song, 2010*). Briefly, rotation and affine transformations were applied to each set of in situ nuclei coordinates to register them individually to the connectome cell nuclei coordinates. The results are presented as a heat map showing for each PR-I the relative frequencies it registered with an in situ centroid of each NT type (*Figure 2e*). To assess the validity of registration, a confusion matrix was constructed (*Stehman, 1997*) (*Figure 2f*; *Supplementary file 1*). In this analysis each set of HCR in situ centroids was registered to all other HCR in situ datasets, and to the connectome centroids. The confusion matrix shows the number of times a registration of the HCR in situ centroid to a connectome centroid corresponds with the registration of another HCR in situ dataset. The higher the values along the diagonal of the matrix, the more the datasets agree with each other when registered to the connectome centroids. From the matrix we observed strong overall support for the registration, although with variable confidence for each photoreceptor. The heat map indicates that among the PR-Is, PR-9 is likely to be exclusively VGAT-positive, while PR-10 is likely to be both VGAT- and VGLUT-positive. The confusion matrix gives high confidence to this assignment, particularly for PR-9. Although PR-9 and PR-10 appear to stand out from the other PR-Is in their NT use, the connectivity of these two photoreceptors in the visuomotor pathway does not appear to be qualitatively different than the other PR-Is (*Figure 2—figure supplement 1*). Finally, the heat map confirms that the other PR-Is are exclusively VGLUT, however with lower confidence for PR-16, which failed to register well.

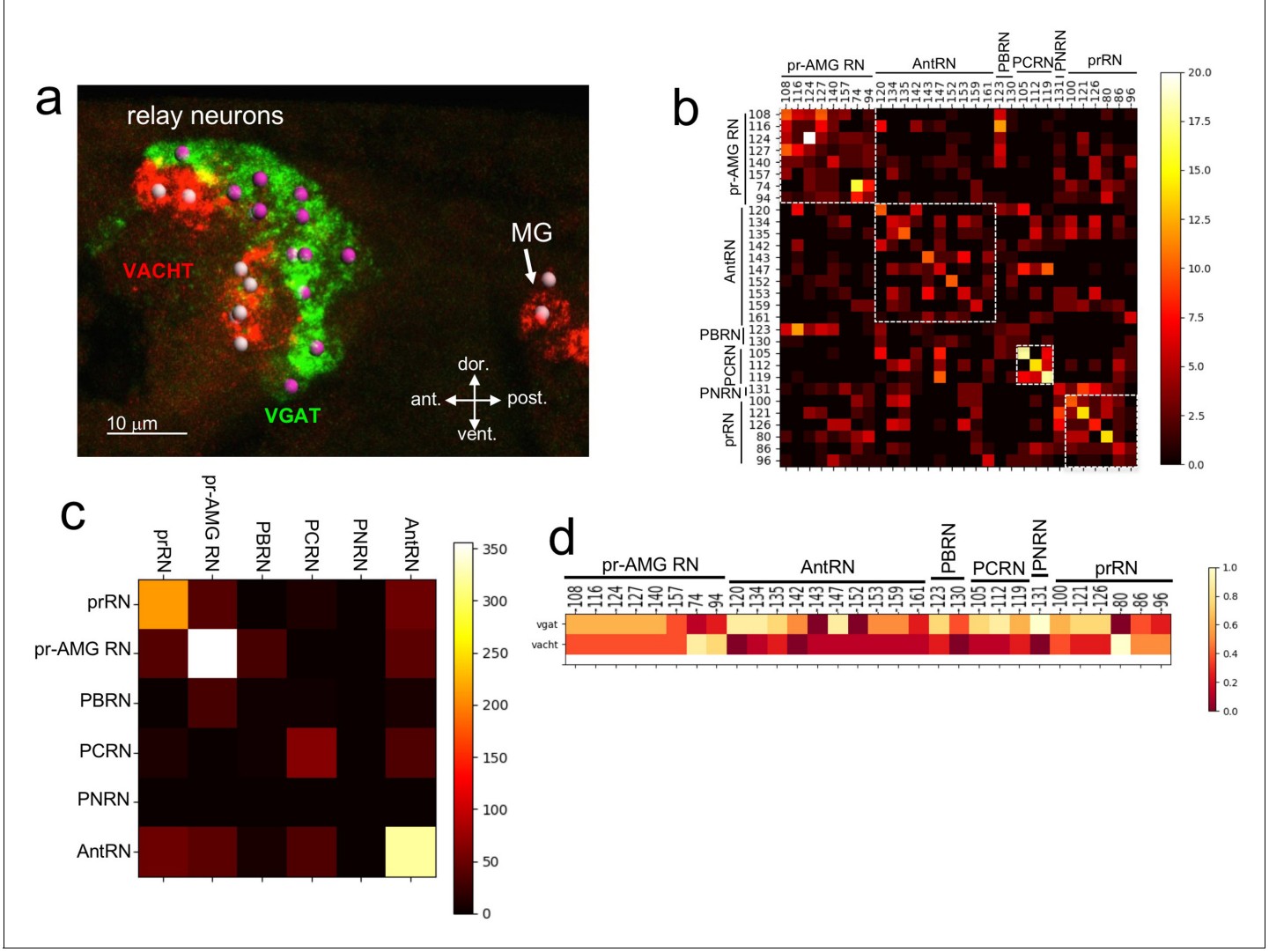

**Figure 3.** Neurotransmitter use in the relay neurons. (**a**) In situ hybridization of VGAT and VACHT to the relay neurons in the brain vesicle. Also visible is the anterior tip of the motor ganglion. Nuclei are shown as spheres. (**b**) Confusion matrix for relay neuron registration. (**c**) Confusion matrix for relay neurons grouped by type. (**d**) Heat map of neurotransmitter predictions from cell registration of relay neurons, with scale showing color by proportion of iterations predicting either VGAT or VACHT. Abbreviations: ant., anterior; post., posterior; dor., dorsal; vent., ventral; MG, motor ganglion; pr-AMG RN, photoreceptor ascending motor ganglion relay neuron; prRN, photoreceptor relay neuron; AntRN, antenna cell relay neuron; PBRN, photoreceptor-bipolar tail neuron relay neuron; PCRN, photoreceptor-coronet relay neuron; PNRN, peripheral relay neuron; VGAT, vesicular GABA transporter; VACHT, vesicular acetylcholine transporter.

DOI: https://doi.org/10.7554/eLife.44753.007

The following figure supplement is available for figure 3:

**Figure supplement 1.** Relay neuron centroids projected in two dimensions.

DOI: https://doi.org/10.7554/eLife.44753.008

## Posterior brain vesicle relay neurons are mixed VGAT- and VACHT-expressing

Sensory input from the photoreceptors, antenna cells, coronet cells, bipolar tail neurons and a sub-set of peripheral neurons is directed to a cluster of ~30 RNs in the pBV. These RNs in turn extend axons through the neck to the MG. Among this cluster are the six prRNs and eight pr-AMG RNs (*Figure 1*; (*Ryan et al., 2016*)). Previous in situ hybridization studies identified VGAT- and VACHT-expressing neurons in the appropriate place in the BV to be RNs (*Yoshida et al., 2004*). Moreover, these neurons project axons posteriorly to the MG, a defining characteristic of the pBV RNs. BV

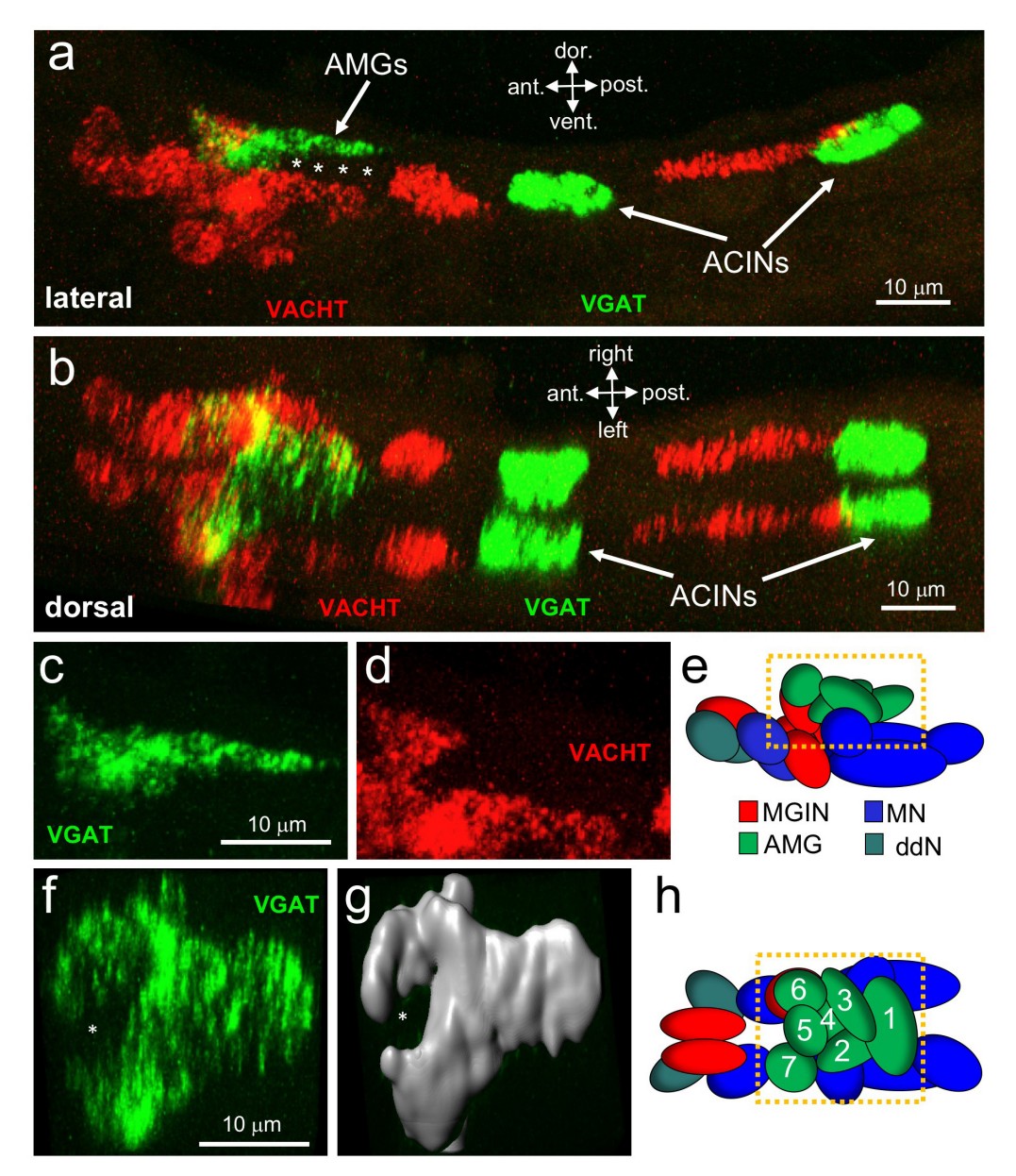

**Figure 4.** Neurotransmitter use in the motor ganglion. (a and b) Expression of VGAT and VACHT by in situ hybridization in the motor ganglion, lateral (a) and dorsal (b) views. Asterisks indicate predicted ependymal cells. (c) Lateral view of VGAT expression in the AMGs. (d) shows same view as c, but with VACHT expression. (e) Diagram of neurons in the motor ganglion (derived from Figure 1 of *Ryan et al., 2017*). Box indicates approximate positions of panels c and d. Lateral view; anterior is to the left. (f) Dorsal view of VGAT expression in the AMGs. Asterisk indicates central non-VGAT expressing cell. (g) Three dimensional surface rendering of VGAT expressing cells in the AMGs. (h) Diagram of a dorsal view of the motor ganglion. AMG cells are numbered. Abbreviations: dor., dorsal; vent., ventral; ant., anterior; post., posterior; AMG, ascending motor ganglion neuron; MGIN, motor ganglion interneuron; ddN, descending decussating neurons; ACIN, ascending contralateral inhibitory neurons; MN, motor neuron; VGAT, vesicular GABA transporter; VACHT, vesicular acetylcholine transporter.

DOI: https://doi.org/10.7554/eLife.44753.009

The following figure supplement is available for figure 4:

**Figure supplement 1.** Representative larvae showing expression pattern for VGAT (green) and VACHT (red) by HCR in situ.
DOI: https://doi.org/10.7554/eLife.44753.010

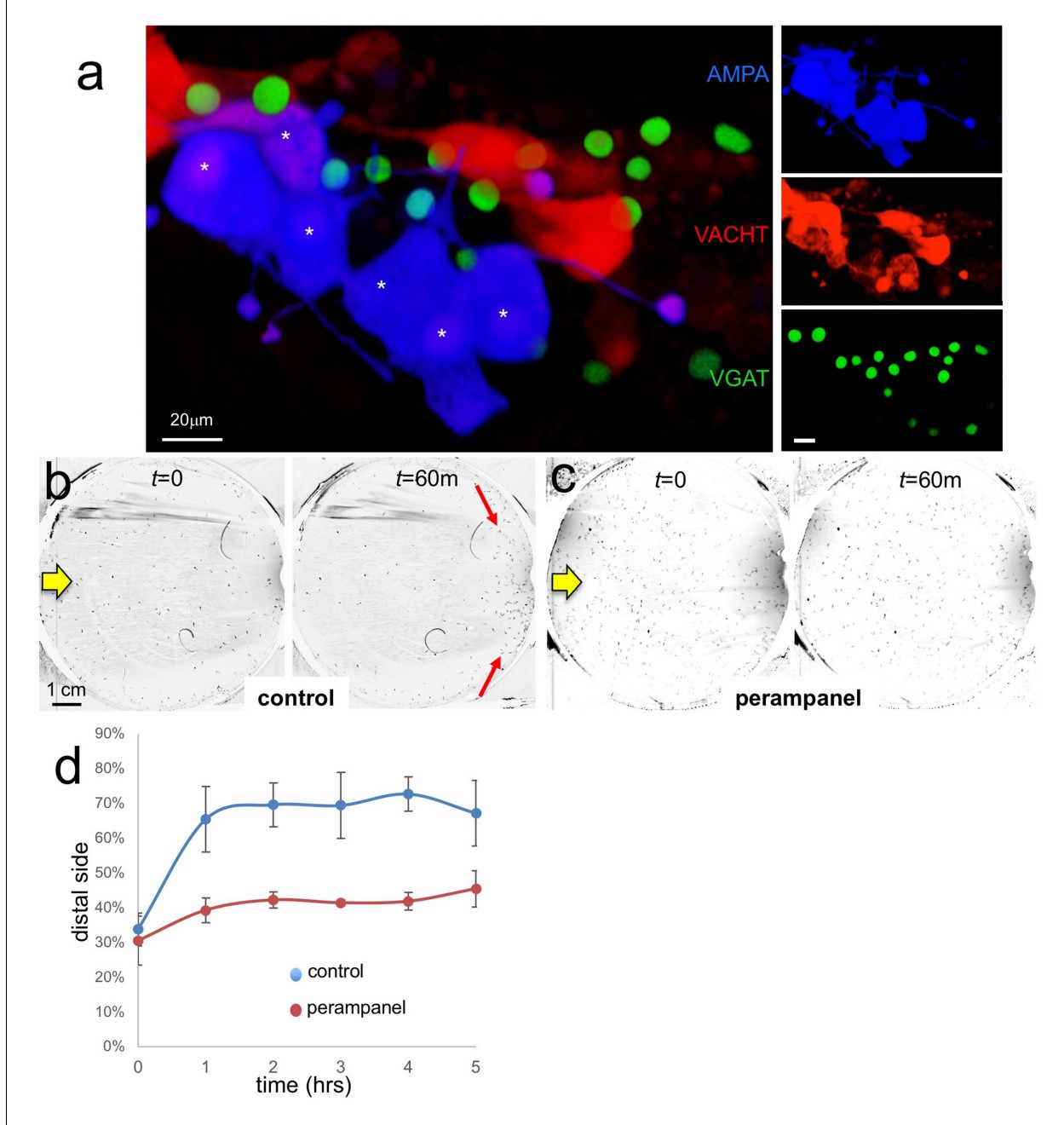

**Figure 5.** AMPA receptors in negative phototaxis. (a) Coexpression of an AMPA-receptor and VACHT expression constructs in the relay neurons (white asterisks). The main panel shows the merge while smaller panels at right show single channels. (b) Negative phototaxis assay in control larvae. Yellow arrow indicates direction of 505 nm light. By 60 min (m) the majority of the larvae have swum to the side of the dish away from the light (red arrow). (c) Perampanel-treated larvae do not show negative phototaxis. (d) Quantification of negative phototaxis in control and perampanel-treated larvae. Points indicate the averages from three independent assays, ±standard deviation. Y-axis represents the percentage of larvae found on the side away from the light source (distal third). Abbreviations: VGAT, vesicular GABA transporter; VACHT, vesicular acetylcholine transporter.

DOI: https://doi.org/10.7554/eLife.44753.011

The following source data and figure supplement are available for figure 5:

**Source data 1.** Source data for *Figure 5D*.
DOI: https://doi.org/10.7554/eLife.44753.013

**Figure supplement 1.** AMPA-receptor neurons in the *Ciona* brain vesicle identified with an AMPA-receptor promoter construct driving GFP.
DOI: https://doi.org/10.7554/eLife.44753.012

neurons expressing other major NTs, including glutamate, dopamine, and serotonin, are neither in the correct brain region to be RNs, nor do they project from the BV to the MG ([*Horie et al., 2008b*; *Moret et al., 2005*; *Pennati et al., 2007*], and our observations). By HCR in situ we observed that the pBV RNs cluster in two distinct groups along the anterior/posterior axis, with the anterior cluster expressing VACHT, and the posterior group expressing VGAT (*Figure 3a*). We observed an average of 16 (±1.6, n = 9 larvae) VGAT-positive neurons and 11 (±1, n = 8 larvae) VACHT-positive neurons.

Unlike the ocellus, the pBV RN cluster does not have obvious anatomical features, although the various classes of RNs are clustered, with, for example, the antenna cell RNs (AntRN) being posterior to the photoreceptor RNs (*Figure 3—figure supplement 1*; *Ryan et al., 2016*). However, given the diversity of RN types in the pBV it is unlikely that the expression domains of VGAT and VACHT precisely correspond to the clusters of RN classes. In order to make predictions of NT use in the RNs we used the same registration approach as with the photoreceptors (n = 7 VGAT/VACHT double in situ datasets, *Figure 3—figure supplement 1*). The confusion matrix for the RNs shows a lower level of convergence than for the PR-Is, suggesting that the cellular anatomy of the RN cluster is less structured than the ocellus (*Figure 3b*; *Figure 3—figure supplement 1*). However, the confusion matrix also shows that the RNs are most often confused for other RNs of the same class (white boxes in *Figure 3b*). This is most evident when the registration is performed not with single cells, but with pooled RNs of each class (*Figure 3c*), and is presumably a reflection of the clustering of RN classes in the pBV. Thus we can have higher confidence in the NT use by RN class than we can have in individual neuron identities. For example, the connectome shows the AntRNs are clustered at the rear of the BV (*Figure 3—figure supplement 1*; *Ryan et al., 2016*)), as are the VGAT expressing neurons (*Figure 3a*; *Figure 3—figure supplement 1*). Accordingly, the registration predicts that eight of the ten AntRNs are VGAT positive (*Figure 3c*). For the present study, which focuses on the visuo-motor pathway, the registration predicts that five of the eight pr-AMG RNs are VGAT expressing, two are VACHT expressing, and one (pr-AMG RN 157) cannot be resolved (no dual VGAT/VACHT expression was observed in the *in situs*). On the other hand, the registration predicts that the six prRNs are evenly mixed between VGAT and VACHT expression. These predictions provide starting points for experimental validation detailed below.

## The motor ganglion contains a mixture of cholinergic and GABAergic neurons

The MG contains five left/right pairs of motor neurons, as well as several classes of interneurons, including six MGINs, seven AMGs, two ddNs, and two posterior MG interneurons (*Ryan et al., 2016*). Also described in the MG are two left/right pairs of decussating VGAT-positive neurons (*Horie et al., 2009*; *Nishino et al., 2010*). These are likely the same decussating MG neurons as described in the connectome, although the names are slightly different (*anterior caudal inhibitory neurons* (*Horie et al., 2009*) versus *ascending contralateral inhibitory neurons* (*Ryan et al., 2016*), both abbreviated as ACIN). However, the connectome reports only three ACINs, with the anterior ACIN not paired. It was speculated that this was an anomalous feature of the particular larva used for the ssEM. Supporting this, a second larva being analyzed by ssEM for connectomics shows two pairs of ACINs (K. Ryan, personal communication).

Like the ocellus, the MG has a well-defined anterior-to-posterior and dorsal-to-ventral cellular anatomy (*Figure 4a and b*; *Ryan et al., 2016*; *Ryan et al., 2018*). Neurotransmitter use by some MG neurons is already documented, including the motor neurons, which are cholinergic (*Takamura et al., 2010*; *Takamura et al., 2002*), and the ACINs which are glycinergic (*Nishino et al., 2010*). By HCR in situ hybridization we observed VGAT- and VACHT-positive neurons in the MG (*Figure 4b*), but no VGLUT- or

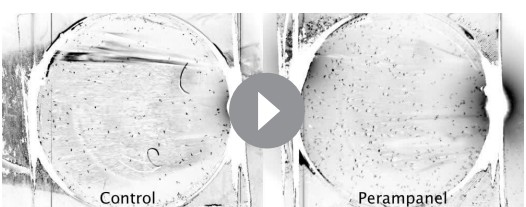

**Video 1.** Negative phototaxis of control and perampanel-treated *Ciona* larvae in 10 cm petri dishes. Directional 505 nm illumination is from the left. Frames were taken at 1 per minute over five hours. In the video the 5 hr is compressed to 15 s (i.e., 1200X normal speed). Black and white tones were inverted to make the larvae more visible.
DOI: https://doi.org/10.7554/eLife.44753.014

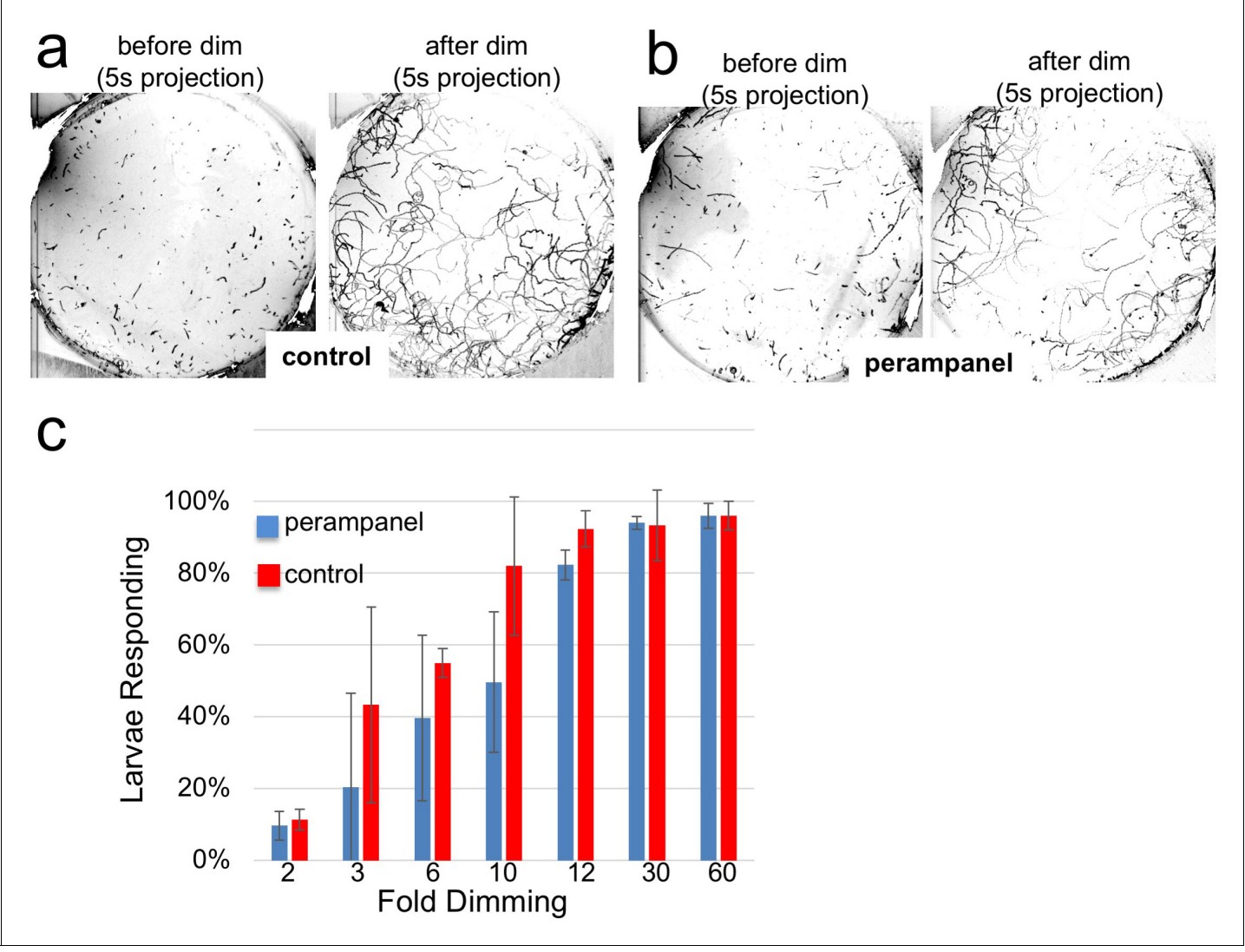

**Figure 6.** Perampanel does not disrupt the light dimming response. (**a**) Light dimming response in control larvae. Shown are 5 s (s) projections from time-lapse videos in which swims appear as lines. Left panel shows a projection 5 s before dimming, and right panel 5 s after dimming. (**b**) same as a, but larvae were perampanel-treated. (**c**) Quantification of light dimming response in control and perampanel treated larvae. Larvae were exposed to dimming of 505 nm light from 2- to 60-fold. Dimming response was scored as percent of larvae responding. Bars show averages of three independent assays ± standard deviation.

DOI: https://doi.org/10.7554/eLife.44753.015

The following source data is available for figure 6:

**Source data 1.** Source data for *Figure 6C*.
DOI: https://doi.org/10.7554/eLife.44753.016

TH-positive cells (data not shown). These results are consistent with previous studies (*Horie et al., 2008b*; *Moret et al., 2005*). Likewise it was reported that no serotonergic cells were present in the MG (*Pennati et al., 2007*). As with the RNs, the VGAT- and VACHT-expressing neurons in the MG are segregated anatomically. We also found a population of 6–7 cells between the AMGs and the MNs (asterisks in *Figure 4a*), that were not annotated in the connectome as neurons and that failed to label with any of our NT markers. We hypothesize that these are ependymal cells, which are abundant in the nerve cord immediately caudal to this region.

Because of the highly structured MG cellular anatomy, we can identify the various MG cell types in the in situ data. The anterior group of VGAT-positive cells is clustered dorsally in the MG, and correspond to AMGs (4 c, d and e; (*Ryan et al., 2017*)). In a dorsal view of the MG (*Figure 4f,g and h*)

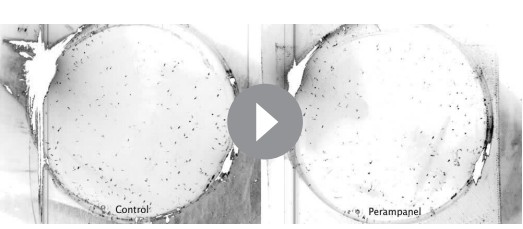

**Video 2.** Swimming of control and perampanel-treated *Ciona* larvae in a directional light field. Larvae in 10 cm petri dishes were recorded at nine frames/second. Black and white tones were inverted to make the larvae more visible. The video plays at 5X normal speed.
DOI: https://doi.org/10.7554/eLife.44753.017

a ring of VGAT-positive cells was observed with a non-VGAT expressing cell in the center (asterisk, *Figure 4f and g*). The VGAT-expressing cells appear to be AMGs 1, 2, 3, 4, 6, and 7, while the central cell, which is instead positive for VACHT, appears to be AMG5. The connectome shows that AMG5 differs in its connectivity from the other AMGs. Significantly, AMG5 is the principle synaptic input for PNS neurons. It then synapses to the other AMGs, which in turn project their axons to other cells in the MG, including MGINs and MNs, as well as to the pr-AMG RNs in the BV. In the posterior of the MG we observed two pairs of VGAT-positive neurons, as described previously (*Horie et al., 2009*). Finally, in the ventral MG we observed a continuous block of VACHT expression that encompasses the anterior three pairs of MNs, the ddNs, and the MGINs. Similar in situ patterns were observed in most larvae (*Figure 4—figure supplement 1*), although the positions of the ACINs were offset in several (see larvae 5 and 6 in *Figure 4—figure supplement 1*), and one larva was observed to be missing both one motor neuron and one ACIN (larva 7in *Figure 4—figure supplement 1*), suggesting that MG variants, such as was observed in the animal used in the connectome study, may be relatively common.

## Parallel visuomotor circuits

Our results indicate that the PR-Is, with the exception of two cells, are glutamatergic, while the PR-IIs are a mixture of GABAergic and GABA/glutamatergic. The *Ciona* genome contains a single glutamate AMPA receptor (AMPAR) (*Okamura et al., 2005*) that is expressed in larvae in the two antenna cells, and in a small cluster of neurons in the pBV (*Hirai et al., 2017*). Published results show that most of the pBV group of AMPAR-positive neurons are clustered at the ends of Arrestin-labeled photoreceptor axons, and that they extend their axons to the MG, suggesting they are photoreceptor RNs (see Figure 2B" in *Hirai et al., 2017*). We find that this pBV group is composed of ~6 cells (*Figure 5—figure supplement 1*). To investigate this further, we co-expressed an pAMPAR >GFP construct (*Hirai et al., 2017*) with pVACHT >CFP and pVGAT >nuclear RFP constructs. We observed coexpression of the AMPAR reporter in a subset of the VACHT-positive RNs, but never in the VGAT-expressing RNs (*Figure 5a*).

To assess the function of the AMPAR-positive cells in *Ciona* visuomotor behaviors we used the non-competitive AMPAR antagonist perampanel (*Hanada et al., 2011*). For the assay, larvae were treated at 25 hr post fertilization (hpf) with perampanel in sea water and compared to vehicle-treated control larvae for both negative phototaxis and response to light dimming. The negative phototaxis assay consisted of placing the larvae in a 10 cm petri dish of sea water with a 505 nm LED lamp placed to one side (described by us previously *Salas et al., 2018*). Images were collected at 1 min intervals over 5 hr to assess for taxis (*Video 1*). *Figure 5b and c* show representative frames from the time-lapse capture at the start and at 60 min for control and perampanel-treated larvae, respectively. In the control sample the larvae at 60 min were observed to cluster at the side of the petri dish away from the light (distal side; red arrows in *Figure 5b*). By contrast no taxis was observed in the perampanel treated larvae (*Figure 5c*). Combined results from three independent assays (n = 129–365 larvae per group) are shown in *Figure 5d* and presented as the percent of larvae found on distal third of the

**Video 3.** Dimming response of control and perampanel-treated *Ciona* larvae in 10 cm petri dishes. Larvae were imaged for 70 s at five frames/second, with dimming of 505 nm ambient light at 10 s. Black and white tones were inverted, and thus the dimming appears as a brightening. The video plays at 5X normal speed.
DOI: https://doi.org/10.7554/eLife.44753.018

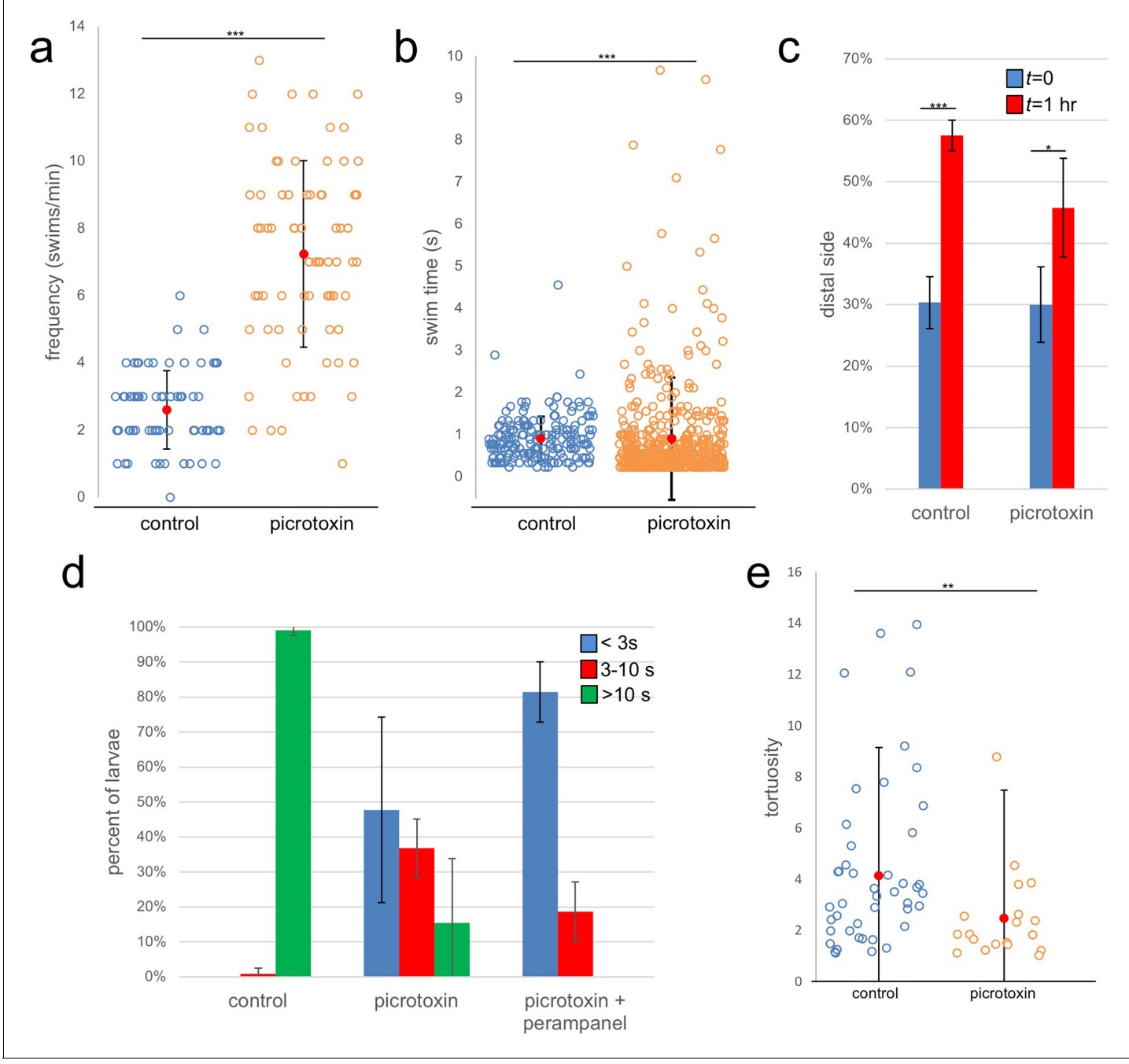

**Figure 7.** Effects of the GABA receptor antagonist picrotoxin on swimming behavior. (a) Frequency of spontaneous swims for control (vehicle only) and picrotoxin-treated larvae in dark conditions (*i.e.*, 700 nm illumination). Each open circle represents a single tracked larva with the number of swim bouts in one minute presented. Also shown are the averages (red circles)±standard deviation. (b) Duration of spontaneous swims in seconds (s). Each circle is one swim bout recorded during a one-minute capture session. (c) Picrotoxin-treated larvae retain negative phototaxis. Bars show the averages of three trails and present the percentage of larvae in the side of the petri dish opposite the illumination (distal third of dish). (d) Dimming response is diminished in picrotoxin-treated larvae, and further diminished by cotreatment with picrotoxin and perampanel. Shown are the averages of three independent trials (37–99 larvae quantified per trial). Data shows the duration in seconds (s) of swims. (e) Tortuosity of sustained swims (>10 s) for control and picrotoxin-treated larvae. Data is presented as in panel a. *, p<0.05; **, p<0.01; ***, p<0.001.
DOI: https://doi.org/10.7554/eLife.44753.019

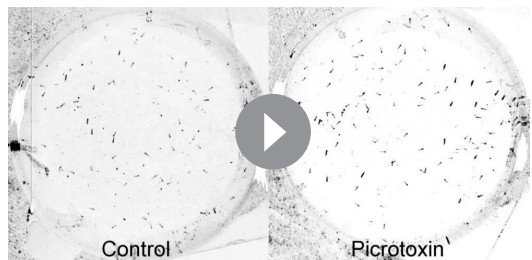

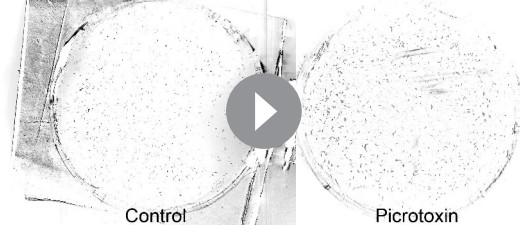

**Video 4.** Spontaneous swimming behavior in control and picrotoxin-treated *Ciona* larvae. Larvae in 6 cm petri dishes were recorded for 1 min at nine frames/second with 700 nm illumination. Black and white tones were inverted to make the larvae more visible. The video plays at 5X normal speed.
DOI: https://doi.org/10.7554/eLife.44753.020

**Video 5.** Negative phototaxis of control and picrotoxin-treated *Ciona* larvae in 10 cm petri dishes. Picrotoxin-treated larvae can be seen to stop swimming at ~1 hr. Directional 505 nm illumination is from the left. Frames were taken at one per minute over five hours. In the video the 5 hr is compressed to 15 s (i.e., 1200X normal speed). Black and white tones were inverted to make the larvae more visible.
DOI: https://doi.org/10.7554/eLife.44753.021

petri dish. For control larvae ~ 70% swam to the distal third within 1 hr, while the perampanel-treated larvae remained evenly distributed across the dish.

The inability of the perampanel-treated larvae to undergo phototaxis was not the result of an inability to swim, as seen in *Video 2* which was taken at 8.9 fps, with and without perampanel. Moreover, we observed that perampanel treatment had no effect on the light dimming response (*Video 3*). *Figure 6a and b* show 5 s projection images from *Video 3* immediately before and after dimming. In these images swims appear as lines, and the responses in control and perampanel-treated larvae appear qualitatively similar. To quantitatively compare dimming response, control and perampanel-treated larvae were exposed to a range of dimming intensities from 2 to 60-fold and the percentage of larvae responding was measured and presented as a percentage in *Figure 6c* (results are from three independent assays, with 46–139 larvae per group). The percentage responding at all intensities was very similar for both groups, and pair-wise comparisons at each fold change failed to show significance. In addition, no differences were measured in the velocity or duration of swims in pair-wise comparisons of control and perampanel-treated larvae at any fold-dimming (data not shown). We conclude that there is no change in sensitivity to dimming caused by perampanel treatment, while phototaxis was completely disrupted. Finally, we also observed that the touch response was not inhibited by perampanel (data not shown), despite the presence of VGLUT-positive epidermal sensory neurons (*Horie et al., 2008b*). This would appear to agree with the observation that primary RNs for the PNS, the eminens cells and the AMGs do not express the AMPAR (*Hirai et al., 2017*; and our observations). In addition to the AMPAR, the *Ciona* genome contains several other glutamate receptors including one kainate and one NMDA (*Okamura et al., 2005*), although their expression has not been characterized.

In summary, we are able to separate the phototaxis and dimming behaviors pharmacologically. Moreover, we can identify the VACHT/AMPAR-positive RNs as essential for an excitatory PR-I circuit that involves presynaptic glutamatergic PR-Is and postsynaptic cholinergic MGINs. The number and location of the VACHT/AMPAR-positive RNs, the circuit logic, and our behavioral observations are all consistent with these being prRNs.

## A disinhibitory circuit

Of equal significance to our observation that navigation is inhibited by perampanel, is our observation that the dimming response, which is mediated by the PR-IIs (*Salas et al., 2018*), is not inhibited by perampanel (*Figure 6*). Our expression studies show that the PR-IIs are comprised of a mixture of VGAT- and VGAT/VGLUT-expressing photoreceptors. Although it is formally possible that PR-IIs signal exclusively via glutamate in an excitatory circuit via a non-AMPA glutamate receptor on their RNs, our observations that several of the PR-IIs are VGAT-only, as are the majority of the pr-AMG RNs, suggests an alternative disinhibitory circuitry logic. This circuit would consist of the inhibitory PR-IIs synapsing to the pr-AMG RNs to reduce their inhibition on the cholinergic MGINs.

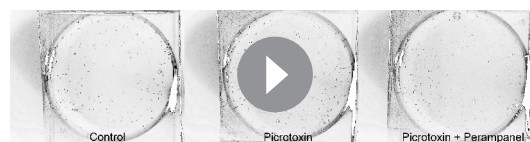

**Video 6.** Dimming response of control and picrotoxin-treated *Ciona* larvae in 10 cm petri dishes. Larvae were imaged for 20 s at five frames/second, with dimming of 505 nm ambient light at 10 s. Black and white tones were inverted, and thus the dimming appears as a brightening. The video plays at 2X normal speed.
DOI: https://doi.org/10.7554/eLife.44753.022

Implicit in the disinhibitory model is an autonomous level of motor activity in larvae that could be inhibited by the GABAergic pr-AMG RNs, and that this inhibition is released upon stimulation of the GABAergic PR-IIs. We investigated this possibility by two approaches. In the first approach, we inhibited GABAergic receptors with picrotoxin (*Olsen, 2014*), which should inhibit signals from the GABAergic photoreceptors and the pr-AMG RNs (and most likely the AntRNs), as well as PNS relay neurons, including the eminens cells and the AMGs. The ACINs, which are essential for the central pattern generator (*Nishino et al., 2010*), are glycinergic and should not be inhibited by picrotoxin. In the second approach, we took advantage of a previously described *Ciona* mutant, *frimousse (frm)* (*Deschet and Smith, 2004*; *Hackley et al., 2013*). In homozygous *frm* larvae the anterior BV is transfated to epidermis due to a null mutation in a neurula stage-specific connexin gene (*Hackley et al., 2013*). *Frm* larvae thus lack the ocellus pigment cell and photoreceptors, as well as the otolith, although the motor ganglion appears intact (*Deschet and Smith, 2004*; *Hackley et al., 2013*).

## Pharmacological GABA receptor inhibition increases spontaneous swims, but decreases dimming response

We first assessed the effects of picrotoxin on spontaneous swimming. As reported previously, when observed under far-red illumination (*i.e.*, outside of the larval response range [*Nakagawa et al., 1999*]) *Ciona* larvae display spontaneous swims consisting primarily of short 'tail flicks', with very few sustained swims (*Salas et al., 2018*). In these conditions, we observed that the frequency of spontaneous swims in the picrotoxin-treated group increased significantly when compared to vehicle-treated (*Figure 7a*; *Video 4*; $p=2.2\times10^{-16}$, n = 75 for both). For this assay, the swimming activity of the larvae was recorded in 1 min videos at 8.9 fps. Each circle in *Figure 7a* corresponds to a single larva tracked over one minute, and the number of swim bouts for each larva during the 1 min is plotted along with the average (red circle) and standard deviation (S.D.). In comparing the duration of the spontaneous swims of the two groups (picrotoxin- and vehicle-treated) we observed an interesting distribution (*Figure 7b*). Overall, the swims for the picrotoxin group were shorter ($p=8.8\times10^{-12}$, n = 184 and 542 for control and picrotoxin respectively), although the picrotoxin group showed more variation, with a number of long-swimming outliers.

In the above assays, behavioral responses were measured within ~20 min of adding picrotoxin (or vehicle only). We observed that longer exposure to picrotoxin (>1 hr) resulted in the nearly complete inhibition of spontaneous and induced swimming behavior, presumably due to overactivation at excitatory synapses following removal of inhibitory input. However, this could be reversed by washing out the picrotoxin (data not shown). This inhibitory effect was evident in assessing negative phototaxis behavior of picrotoxin-treated larvae. While these assays are typically conducted over several hours (*e.g.*, *Figure 5d*), this was not possible with the picrotoxin-treated larvae. Nevertheless, the picrotoxin-treated larvae did show negative phototaxis when measured at 1 hr, although the response was dampened in comparison to controls (*Figure 7c*; $p=8.7\times10^{-4}$ and p=0.03 for control and picrotoxin respectively, n = 137–487; also see *Video 5*).

When assessed for the dimming response the picrotoxin-treated larvae showed a large decrease in evoked sustained swims (defined as lasting longer than 10 s) (*Figure 7d*; *Video 6*). Instead, we observed a preponderance of very short swims that likely reflect the elevated rate of spontaneous tail-flick swims in the larvae. Nevertheless, there remained a number of sustained swims in the picrotoxin-treated larvae. While it is possible that these were due to incomplete inhibition by picrotoxin, we feel it was more likely due to a small contribution from the PR-Is to the dimming response, as we have speculated previously (*Salas et al., 2018*). In support of this, we observed that treatment of larvae with picrotoxin and perampanel completely eliminated the evoked sustained swims. We documented that swims evoked by the PR-Is have lower tortuosity (*i.e.*, are straighter) than those evoked

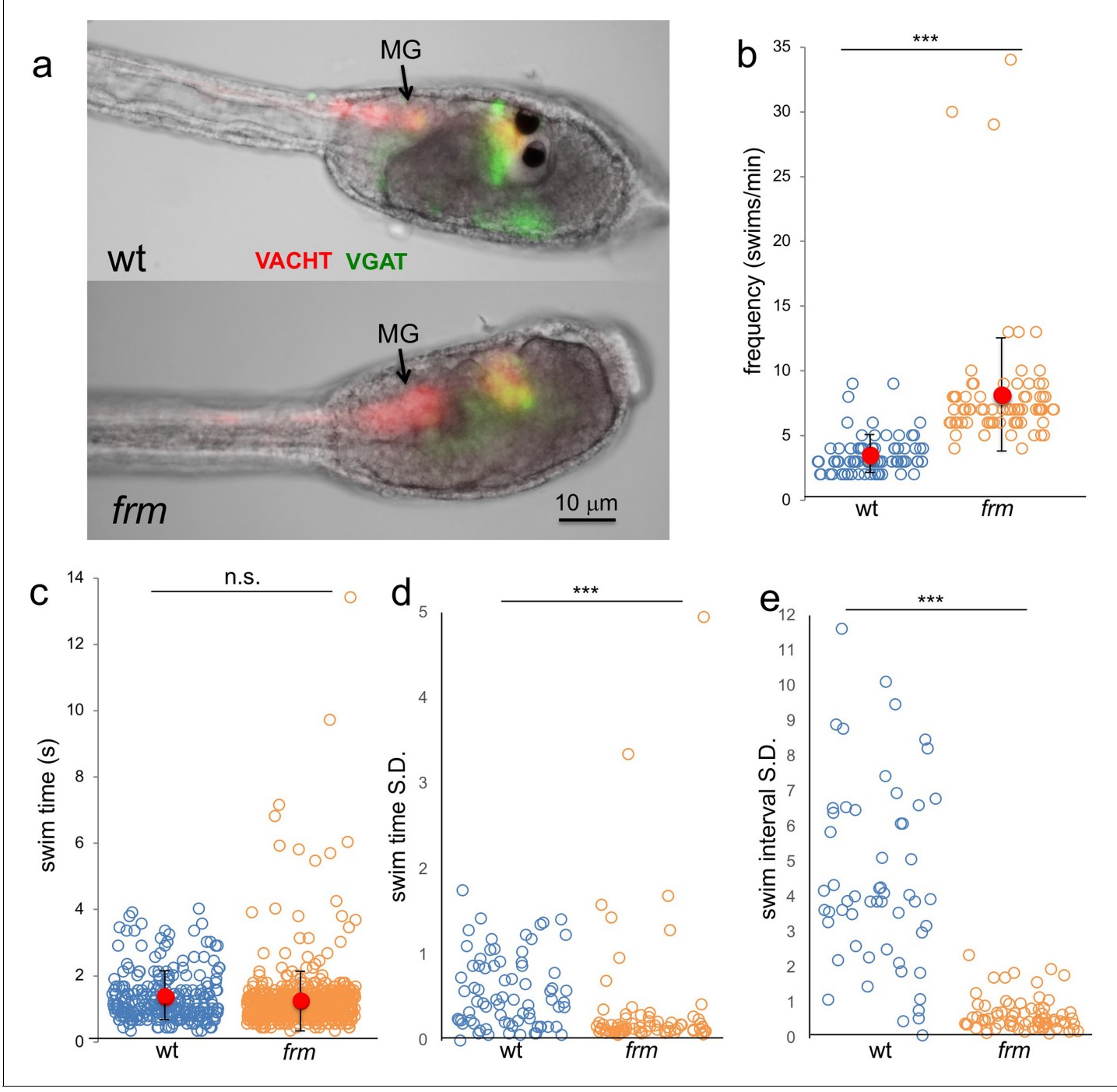

**Figure 8.** Behavior of homozygous *frimousse* (*frm*) larvae. (a) VGAT and VACHT reporter construct expression in wild type (wt) and frm larvae. (b) Frequency of spontaneous swims of wt and frm larvae in dark conditions (i.e., 700 nm illumination). Each open circle represents one tracked larva with the number of swim bouts in one minute presented. Also shown are the averages (red circles) ±standard deviation. (c) Duration of all spontaneous swims in one-minute recording presented in seconds (s) for wt and frm larvae. (d) Standard deviation (S.D.) of the duration of swim bouts over one minute for each individual larva recorded. (e) S.D. of the interval between swim bouts over one minute for each individual larva recorded. For d and e, larvae with <4 swim bouts were not included in the analysis. ***, p<0.001; n.s., not significant.

DOI: https://doi.org/10.7554/eLife.44753.023

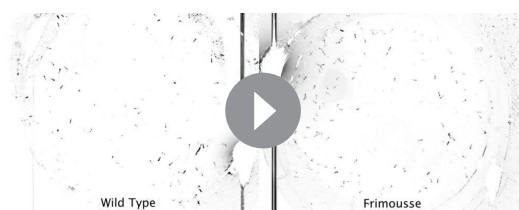

**Video 7.** Spontaneous swimming behavior in wild type and homozygous frimousse larvae. Larvae in 6 cm petri dishes were recorded for 1 min at 9 frames/second with 700 nm illumination. Black and white tones were inverted to make the larvae more visible. The video plays at 5X normal speed.

DOI: https://doi.org/10.7554/eLife.44753.024

by the PR-IIs (*Salas et al., 2018*). Consistent with this, we found the tortuosity of the sustained swims from the picrotoxin-treated larvae had lower tortuosity than those of the vehicle-treated controls (*Figure 7e*; n = 19 and 60, respectively; p=0.003).

## Frimousse mutants have increased spontaneous swim frequency

We found that observation of larvae homozygous for the *frm* mutation also supported the disinhibitory mechanism. As would be predicted due to their loss of photoreceptors (*Hackley et al., 2013*), *frm* larvae showed no response to light (our unpublished observation). Despite the defects in the anterior BV, the MG is intact in *frm* larvae as assessed by both gene expression and morphology (*Deschet and Smith, 2004* and *Figure 8a*). Moreover, not only can *frm* larvae swim, they show increased frequency of spontaneous swims compared to wild type larvae (*Figure 8b*) when assessed using the same parameters as for the picrotoxin-treated larvae (n = 75 for both; p<$5\times10^{-16}$). Although the frequency of swims was higher in *frm* larvae, the average swim time was not significantly different between the two (*Figure 8c*; n = 260 and 608, respectively). However as with the picrotoxin-treated larvae, a handful of very long swims were observed uniquely in the *frm* group (*Figure 8c*). Despite the similarity in the swim times between the *frm* and wild type larvae, the swim characteristics were very different (*Video 7*), with the swimming of *frm* larvae being much more stereotyped. For example, while average swim times of wild type and *frm* larvae were very similar (*Figure 8c*), the standard deviations of swim times calculated and plotted for each larva show much lower swim-to-swim variation in the *frm* larvae (*Figure 8d*; p<$5\times10^{-15}$). The stereotypy was even more pronounced when the time interval between swims was analyzed (*Figure 8e*). For wild type larvae the standard deviations of interval times showed a wide range of values (*i.e.*, high variability of interval times), while the standard deviations for *frm* larvae were much lower (p<0.0005). We did not observe an increase in stereotypic behavior in comparing picrotoxin- to vehicle-treated larvae (data not shown). The behavior of *frm* larvae is characteristic of an oscillator that evokes spontaneous swims with the frequency of ~8/min. Thus, sensory input from the BV appears to suppress this oscillatory behavior, leading to less frequent and more varied swims in wt larvae, supporting a disinhibitory circuit. Interestingly, we observed some VGAT and VACHT expression in the remnant of the *frm* BV (*Figure 8a*). While these expressing cells may be RNs, it remains to be determined whether in the absence of sensory input they develop properly, and if they are functional, how their apparent opposing activities might influence spontaneous swimming (albeit elevated).

## Discussion

*Figure 9* presents a model of the *Ciona* visuomotor circuitry that takes into account the connectome, neurotransmitter use, and behavioral observations. Absent from this model is the detailed and unique connectivity of each neuron in these pathways (*Figure 1—figure supplement 1*), as well as the inputs from other neurons which are not part of the minimal circuit. Nevertheless, we feel that this model will serve as a useful starting point for more detailed analyses of these components. Our findings support a model for two parallel visuomotor pathways, one mediated by the PR-Is and sensitive to the direction of light, and the other mediated by the PR-IIs and sensitive to changes in ambient light. A number of other sensory systems, including mammalian vision and olfaction and *Drosophila* $CO_2$ detection (*Callaway, 2005*; *Geramita et al., 2016*; *Lin et al., 2013*) similarly split components of sensory information into parallel circuits. The PR-I circuit is a simple excitatory pathway with glutamatergic photoreceptors projecting to cholinergic prRNs, exciting them via cation-specific ionotropic AMPARs. The prRNs in turn synapse to the cholinergic MGINs, and then these onto the MNs. The fact that glutamate is used by the *Ciona* larvae exclusively in sensory neurons

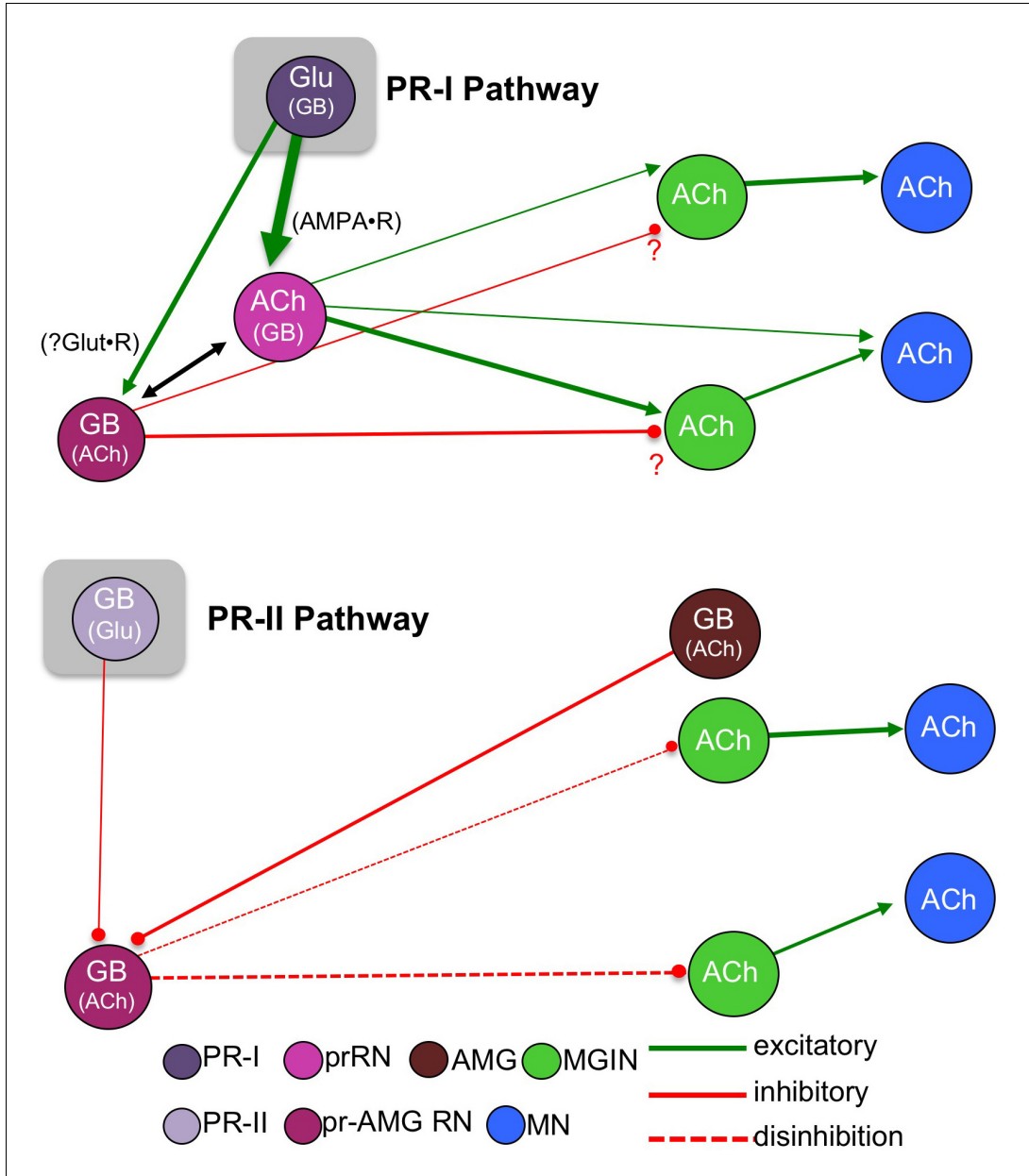

**Figure 9.** Models showing parallel visuomotor pathways for negative phototaxis (top) and light dimming response (bottom). Neurotransmitters in parentheses are thought to play a lesser role in the proposed pathway. Abbreviations: PR-II, photoreceptor group II; PR-I, photoreceptor group I; pr-AMG RN, photoreceptor ascending motor ganglion relay neuron; prRN, photoreceptor relay neuron; MGIN, motor ganglion interneuron; MN, motor neuron; Glu, glutamate; GB, GABA; ACh, acetylcholine. Cell types are color coded according to *Satoh (1994)*.
DOI: https://doi.org/10.7554/eLife.44753.025

(photoreceptors, antenna cells, and epidermal sensory neurons), coupled with the very limited distribution of AMPARs, allowed us to validate essential components of this circuitry with perampanel. The PR-Is also synapse onto the pr-AMG RNs, which are predicted to be primarily GABAergic. Our observation that AMPAR expression is exclusive to the cholinergic RNs suggests that the response of GABAergic cells to the PR-Is may differ from cholinergic cells, and perhaps plays a role in visual information processing. In fact, the interconnections between the pr-AMG RNs and the AMPAR-expressing prRNs (black arrow *Figure 9*; see also *Figure 1—figure supplement 1*), are suggestive of an incoherent feedforward loop (*Alon, 2007*). We have already documented that *Ciona* larvae are

able to phototax in a wide range of illumination conditions (*Salas et al., 2018*), and moreover, we have found that *Ciona* larvae show robust fold-change detection (*Adler and Alon, 2018*) behavior (manuscript in preparation). Together these observations suggest that the RN cluster plays a role in visual processing, rather than simply passing information to the MG.

Our model for the PR-II mediated dimming/escape behavior is more surprising, and includes a novelty – inhibitory photoreceptors. From in situ hybridization we observed that some PR-IIs exclusively express VGAT, while other co-express VGAT and VGLUT. The significance of VGAT/VGLUT coexpression in the *Ciona* visuomotor pathway is not yet clear, although similar coexpression is widely observed in mammalian brains (*Fattorini et al., 2015*; *Zander et al., 2010*) and invertebrates (*Fabian-Fine et al., 2015*). It is speculated that co-release of GABA and GLUT may serve to tune excitatory/inhibitory balance. While the connectome shows that not all of the PR-IIs project to the RNs, with a subset instead forming extensive connections to other PR-Is and PR-IIs, the connectome indicates that several of the VGAT-exclusive PR-IIs do project to the pr-AMG RNs (*Figure 2e* and *Ryan et al., 2016*), consistent with our hypothesis that the PR-II output to the pr-AMG RNs is predominantly inhibitory.

The *Ciona* genome encodes seven ionotropic/Cl$^-$ GABA receptor subunit genes (GABA$_A$) (*Okamura et al., 2005*), but does not have an ortholog of the cationic EXP-1 GABA receptor (*Beg and Jorgensen, 2003*), confirmation that the GABAergic synaptic events are most likely inhibitory. In addition, electrophysiological studies done nearly fifty years ago on larvae of the ascidian *Amaroucium constellatum* reported that their photoreceptors, like those of vertebrates, were hyperpolarizing (*Gorman et al., 1971*). In other words, dimming is likely to result in a release of GABA from the PR-IIs. Although the heterogeneity of ascidian photoreceptors (*e.g.*, PR-I and –II) was not known at the time, both the vertebrate-like ciliary structure shared by all *Ciona* photoreceptors and the structure of *Ciona* opsins appear to rule out the possibility of depolarizing phototransduction (*Kusakabe and Tsuda, 2007*; *Kusakabe et al., 2001*). Also in agreement with an inhibitory output from the PR-IIs is our prediction that the majority of the pr-AMG RNs, the exclusive RNs of the PR-IIs, are themselves GABAergic, which would make a disinhibitory circuit most plausible (*Figure 9*). We also show that removal of BV sensory input with the *frm* mutant, or inhibition of GABA receptors with picrotoxin, leads to more frequent spontaneous swims, suggesting that a disinhibitory pathway could lead to stimulation of swimming. Finally, we observed that the AMGs, with the exception of one cell, are GABAergic. The AMGs are one of the primary relay centers for the PNS (*Ryan et al., 2018*) and project to the MGINs and MNs. However, the AMGs also project ascending axons to the pr-AMG RNs. It is thought that the convergence of PR-II and AMG inputs at the pr-AMG RNs serves to initiate an integrated escape response (*Ryan et al., 2018*). Our finding that these two classes of neurons (PR-IIs and AMGs) are likely to have the same input (inhibition) on the pr-AMG RNs further bolsters the integrated response model. Finally, the PR-II mediated dimming response was not inhibited by the AMPAR antagonist perampanel, suggesting the PR-II glutamate release at pr-AMG RNs acts through other receptors, such as the NMDA receptor, and may be more involved in modulating or processing the visual response, and that GABA release may be more important.

Validation of this hypothetical disinhibitory circuit will require analysis of individual neurons in behaving larvae. Although we have been able to get robust GCaMP imagery from the CNSs of transgenic *Ciona* larvae (our unpublished observations), the fact that the excitation and emission spectra of GCaMP (as well as red-shifted calcium indicators) overlap with the behavioral spectrum of *Ciona*, and the inefficacy of GCaMP for visualizing inhibition, led us to abandon this approach. We are currently exploring methods for electrophysiological recording of *Ciona* BV neurons.

## Ascidians and the evolution of vertebrate visual systems

The evolutionary relationship between the ascidian ocellus and the visual organs of cephalochordates (*e.g.*, amphioxus) and vertebrates remains unclear (*Kusakabe and Tsuda, 2007*; *Lamb et al., 2007*; *Lamb, 2013*). The observations that the *Ciona* PR-I and PR-II complexes are distinct morphologically, mediate different behaviors, project via distinct visuomotor circuits, and express different NTs, raises the possibility that these two complexes may have independent origins, and thus have different evolutionary relationships to the photoreceptor organs of other chordates. We speculate that the ascidian PR-I complex is likely to be homologous to the vertebrate lateral eyes and the amphioxus frontal eye, which like the PR-I complex is pigmented and appears to play a role in detecting the direction of light, although not necessarily in taxis (*Stokes and Holland, 1995*). On

the other hand, the pineal eyes of amphibian tadpoles and fish larvae mediate a shadow/dimming response, suggesting homology with the ascidian PR-II photoreceptor complex (*Jamieson and Roberts, 2000*; *Yoshizawa and Jeffery, 2008*). Nevertheless, the inhibitory nature of the *Ciona* PR-IIs makes assigning homologies more difficult. It is possible that use of GABA by these photoreceptors is a derived feature of ascidians, as inhibitory photoreceptors have yet to be described elsewhere. Alternatively, in the vertebrate retina GABAergic/glycinergic horizontal and amacrine cells are prevalent, and, moreover, it has been proposed that these cells, as well as ganglion cells, are derived from an ancient photoreceptor (*Lamb, 2013*; *Arendt, 2003*). While this may imply an alternative evolutionary origin for the *Ciona* PR-IIs, these observations may simply support the plasticity of NT use in visual systems.

# Materials and methods

**Key resources table**

| Reagent type (species) or resource | Designation | Source or reference | Identifiers | Additional information |
|---|---|---|---|---|
| Chemical compound, drug | fluorescently-labeled RNA probes for VGAT, VGLUT, VACHT | Molecular instruments | | |
| Chemical compound, drug | SlowFade Gold | Invitrogen | | |
| Chemical compound, drug | perampanel | Santa Cruz Biotech; Adooq Bioscience | | |
| Chemical compound, drug | picrotoxin | Tocris | | |
| Equipment | 700 nm and 505 nm led light sources | Mightex | | |
| Equipment | light meter | Extech Instruments | | |
| Recombinant DNA reagent | *Ciona robusta* pVGAT > CFP | Yasunori Sasakura, University of Tsukuba, Japan | | |
| Recombinant DNA reagent | *C. robusta* opsin promoter (as pSP-Ci-opsin1) | Takeo Horie, Tsukuba University, Japan; Takehiro Kusakabe, Konan University, Japan | | |
| Recombinant DNA reagent | synthesized RFP | Gene Block; IDT | | |
| Recombinant DNA reagent | *C. robusta* pAMPAR > GFP | Haruo Okado, Tokyo Metropolitan Institute of Medical Science, Japan | | See Hirai et al., (*Hirai et al., 2017*). |
| Recombinant DNA reagent | *C. robusta* pOpsin > RFP | This paper. | | This construct was created using the opsin promoter (Horie and Kusakabe) and a synthesized RFP (Gene Block, IDT). |
| Recombinant DNA reagent | *C. robusta* pVGAT > H2B::RFP | This paper. | | This construct was created using the Gateway cloning system, cloning pVGAT into pDONR-221-P3-P5, then recombining with an H2B::RFP entry clone. |
| Software, algorithm | Python v 2.7 | Python Software Foundation | | |
| Software, algorithm | Microsoft Excel | Microsoft | | |
| Software, algorithm | Imaris v 9.1 | Bitplane | | |
| Software, algorithm | MATLAB | WolframAlpha | | |

*Continued on next page*

*Continued*

| Reagent type (species) or resource | Designation | Source or reference | Identifiers | Additional information |
|---|---|---|---|---|
| Software, algorithm | ELIANE script for MATLAB | This paper. | | Estimator of Locomotion Iterations for Animal Experiments; a MATLAB script for tracking swimming larvae during behavioral assays. |
| Strain, strain background | *C. intestinalis* | Marine Biological Laboratory, Woods Hole, Massachusetts, USA | | Formerly known as *C. intestinalis* type B. |
| Strain, strain background | *C. robusta* | Santa Barbara Harbor, USA; S. Lepage, M-REP, Carlsbad, USA | | Formerly known as *C. intestinalis* type A. |
| Strain, strain background | pVGAT > kaede, *C. robusta* stable line | National Bioresource Project, Japan | | |

## Animals

*Ciona robusta* (a.k.a., *Ciona intestinalis* type A) were collected from the Santa Barbara Yacht harbor or were obtained from M-REP (Carlsbad). *Ciona intestinalis* (type B) were obtained from Marine Biological Laboratory (Woods Hole). The mutant *frimousse* (*frm*) and the pVGAT > kaede stable transgenic line (National Bioresource Project, Japan) were cultured at the UC Santa Barbara Marine Lab, as described previously (*Veeman et al., 2011*). Larvae were obtained by mixing dissected gametes of 3 adults and cultured in natural seawater at 18°C. Homozygous *frm* larvae were produced by natural spawning of heterozygote *frm* adults.

## Transgene Constructs

*pOpsin1 > RFP*. Starting with the plasmid pSP-Ci-opsin 1 (2Kb)>kaede (Takeo Horie and Takehiro Kusakabe, unpublished), the kaede reading frame was replaced with a synthesized RFP (GeneBlock; IDT). *pVGAT > H2B::RFP*. The promoter region of VGAT was amplified from genomic DNA using primers containing adaptors for Gateway cloning attB3 and attB5 sites (ataaagtaggctatttaaacaaccagattgcttctgtct and caaaagttgggt tgaggtcgaacgttccg) (*Yoshida et al., 2004*). This was cloned into pDONR-221-P3-P5 and recombined with an entry clone containing H2B::RFP (*Roure et al., 2007*).

## Transgenesis

### Microinjection

Fertilized one-cell *Ciona intestinalis* (type B) embryos were microinjected through the chorion, as described previously for *C. savignyi* (*Deschet et al., 2003*).

### Electroporation

Unfertilized *Ciona robusta* eggs were dechorionated using 0.1% trypsin in 10 mM TAPS pH 8.2 in filtered sea water. Eggs were then fertilized and electroporated (*Zeller, 2018*) with 40 µg each of pVACHT >CFP (*Horie et al., 2011*) and pVGAT >H2B::RFP. Embryos were cultured at 18°C in filtered sea water with antibiotics until 18 hpf. Larvae were live-mounted for microscopy.

## Hybridization chain reaction (HCR) in situ

*Ciona intestinalis*-type B were used for in situ studies and staged to match the animals used in the connectome study (*Ryan et al., 2016*). Optimized HCR in situ probes for each target transcript were obtained from Molecular Technologies. For detection of GABAergic/glycinergic cells, probes were made to the vesicular GABA transporter gene; for glutamatergic cells, probes were made to the vesicular glutamate transporter for cholinergic cells, probes were made to the vesicular acetylcholine transporter. The sequences from which the HCR probe sets were chosen were assembled from scaffold reads available through the Aniseed website (aniseed.cnrs.fr), and are shown in *Supplementary file 1*. The in situ protocol followed the previously published *Ciona in situ* hybridization protocol (*Corbo et al., 1997*) until the prehybridization step. At this point, the protocol follows

the published HCR protocol (*Choi et al., 2018*), with the following exception: during the amplification stage, incubation with hairpins is performed for 3 days instead of 12–16 hr.

HCR in situ stained larvae were cleared with Slowfade Gold with DAPI (Invitrogen) and imaged on a Leica SP8 resonant scanning confocal microscope. Imaris v. 9.1 (Bitplane) was used to visualize embryos and assign centroids to nuclei using the 'add new spots' function, followed by manual correction when necessary. Nuclei were assigned using the maximum intensity projection, cropped to the area of interest. Volume rendering of in situ patterns was also done using Imaris v. 9.1.

## Cell registration

A rotation matrix was calculated based on the 3-dimensional vectors between the anchor cells (ddN and/or antenna cells) and the center of the target cells (photoreceptors or relay neurons) using the HCR in situ (target set) and connectome cell centroids (source set). The source set was then rotated to an approximate orientation to the target set. Next, the Coherent Point Drift Algorithm was used to calculate an affine transformation matrix between the source set and the target set of cells (*Myronenko and Song, 2010*). This algorithm models the source set as a Gaussian Mixture Model (GMM), and the target set is treated as observations from the GMM. The transformation matrix is calculated to maximize the Maximum A Posteriori estimation that the observed point cloud is drawn from the GMM. A nearest neighbor mapping based on Euclidean distance is then used to find the closest corresponding point in the target cell set for each cell in the transformed source cell set. The implementation used was adapted from the pure Python implementation https://github.com/sia-vashk/pycpd. The maximum number of iterations was set to 1000 and the maximum root mean squared error for convergence was set to 0.001. The code for the registration is available as supplementary material (*Source codes 1–3*).

### Confusion matrix

Each dataset containing NT information was registered to every other dataset of the same type using the algorithm detailed above. The EM-registration based cell assignments of each cell in both sets is then compared to each other to see if they agree (*Stehman, 1997*). The confusion matrix shows the number of times a cell assignment in one dataset corresponds with each other cell assignment in another dataset.

## Behavioral assays

For time-lapse videos the inverted lid of a 60 mm petri dish was first coated with a thin layer of 1% agarose. Larvae were then added to the inverted lid with filtered sea water containing 0.1% BSA with streptomycin and kanamycin each at 20 µg/ml. Finally the dish was covered with a square of glass leaving no air at the top interface. Stock solutions of perampanel were dissolved in methanol and diluted to final concentrations of either 5 µm (Santa Cruz Biotech) or 15 µM (Adooq Bioscience) in filtered sea water/BSA/antibiotics. Picrotoxin (Tocris) was also diluted in methanol and used at a final concentration of 1 mM. Control samples received methanol alone.

Time-lapse images were collected using a Hamamatsu Orca-ER camera fitted on a Navitar 7000 macro zoom lens. Programmable 700 nm and 505 nm LED lamps were used to illuminate the larvae (Mightex). All light intensity readings were taken with an Extech Instruments light meter.

### Dimming-response

All larvae used were between 25 and 28 hpf (18°C). For image capture, the larvae were illuminated with the 700 nm LED lamp and the camera was fitted with a red filter to block the 505 nm light. The videos were recorded at five fps. In the assays, larvae were first recorded for 10 s with the 505 nm LED light mounted above the dish at 600 lux and then dimmed to specific values while image capture continued for another 3 min. Larvae were allowed to recover for 5 min before being assayed again.

### Phototaxis

All larvae used were approximately 25 hpf (18°C). The 505 nm LED light was mounted to one side to the petri dish at approximately 3000 lux. Images were captured at one frame per minute for five hours, with the exception of 30 s capture session at 8.9 fps to assay swimming behavior.

## Spontaneous Swims

All larvae used were between 26 and 28 hpf. The plates were illuminated with only a 700 nm LED light in order to record dark conditions. The videos were recorded at about 8.9 fps for one minute.

## Behavioral data analysis

### Dim-response criteria

Responses to light dimming were counted if: (1) the larva was stationary at the time of the light dimming, and (2) it swam for longer than 3 s. Three seconds was determined by measuring the duration of tail flicks as previously described (*Salas et al., 2018*). Larvae that bumped or brushed against other larvae or the dish edges were not counted.

### Tracking and quantification

Larval swims were tracked using a custom MATLAB script named Estimators of Locomotion Iterations for Animal Experiments (ELIANE). Before uploading to ELIANE, time-lapse images were first processed with Fiji (ImageJ) by subtracting a minimum Z-projection to all the frames and then inverting black and white. ELIANE takes the processed time-lapse images and first creates a background image by averaging the pixels from all the frames. Next, it goes to the initial frame, subtracts the background image, and stores all remaining objects found in the specified region of interest (ROI) as initial objects. Then, analyzing one-by-one the initial objects, it goes frame-by-frame subtracting the background image and analyzing all objects to determine the new position of the object by comparing the Euclidean distances of it to all other objects in that frame. If the object had moved unrealistically fast (>6.5 mm/s), moved outside the ROI, or did not move after a set time (1 min), the object was not analyzed. This MATLAB script can be found in the Supplemental Materials (*Source code 4*).

The spontaneous swims in the *frimousse* experiment were quantified manually.

### Sampling

Assessment of larval swim parameters were performed using three independent assays. For the spontaneous swims, which were quantified manually, 25 larvae were selected randomly, starting from the center of the plate going outward, only using the ones that could be tracked for the entire minute recording session.

### Tests of significance

Dimming response significance and swim frequency were calculated using the Wilcoxon rank-sum test; spontaneous swim time significance was calculated using the Student's *t*-test; and the variance of spontaneous swimming significance was calculated using the F-test.

# Acknowledgements

We thank Takeo Horie and Takahiro Kusakabe for the opsin1 promoter construct; Yasunori Sasakura for the stable pVGAT >kaede line and pVACHT >CFP plasmid; Haruo Okado for the pAMPAR >GFP construct. Kerrianne Ryan for her helpful discussion and sharing unpublished data. Chelsea Parlett-Pelleriti for her advice on statistical analysis. We acknowledge the use of the NRI-MCDB Microscopy Facility and the Resonant Scanning Confocal supported by NSF MRI grant 1625770. This work supported by an award from NIH (NS103774) to WCS and BM.

# Additional information

### Funding

| Funder | Grant reference number | Author |
| --- | --- | --- |
| National Institute of Neurological Disorders and Stroke | R01NS103774 | William C Smith |

The funders had no role in study design, data collection and interpretation, or the decision to submit the work for publication.

## Author contributions
Matthew J Kourakis, Data curation, Formal analysis, Supervision, Investigation, Methodology, Writing—review and editing; Cezar Borba, Software, Formal analysis, Investigation, Methodology, Writing—review and editing; Angela Zhang, Software, Formal analysis, Methodology, Writing—review and editing; Erin Newman-Smith, Conceptualization, Formal analysis, Investigation, Methodology, Writing—review and editing; Priscilla Salas, Investigation, Writing—review and editing; B Manjunath, Conceptualization, Supervision, Funding acquisition, Project administration, Writing—review and editing; William C Smith, Conceptualization, Funding acquisition, Writing—original draft, Project administration

## Author ORCIDs
Matthew J Kourakis https://orcid.org/0000-0002-1261-3811
William C Smith https://orcid.org/0000-0002-6257-7695

## Decision letter and Author response
Decision letter https://doi.org/10.7554/eLife.44753.036
Author response https://doi.org/10.7554/eLife.44753.037

# Additional files

## Supplementary files
• Source code 1. Calculates transformation between each dataset in a folder and the EM data based on nuclei location in 3D.
DOI: https://doi.org/10.7554/eLife.44753.026

• Source code 2. Calculates cell assignments based on an affine transformation between each dataset and the EM data based on nuclei location in 3D.
DOI: https://doi.org/10.7554/eLife.44753.027

• Source code 3. Plots the nucleus locations in 3D for each dataset along with the EM dataset using Plot.ly.
DOI: https://doi.org/10.7554/eLife.44753.028

• Source code 4. Estimator of locomotion iterations for animal experiments: Stimulus Response Tracker v0.07 (Matlab script).
DOI: https://doi.org/10.7554/eLife.44753.029

• Source data 1. Photoreceptor matrix.
DOI: https://doi.org/10.7554/eLife.44753.030

• Source data 2. Relay neuron matrix.
DOI: https://doi.org/10.7554/eLife.44753.031

• Source data 3. Grouped relay neuron matrix.
DOI: https://doi.org/10.7554/eLife.44753.032

• Supplementary file 1. Nucleotide sequences for HCR probe selection.
DOI: https://doi.org/10.7554/eLife.44753.033

• Transparent reporting form
DOI: https://doi.org/10.7554/eLife.44753.034

## Data availability
All data generated or analysed during this study are included in the manuscript and supporting files. Source data files have been provided for Figures 1, 2, 5 and 6.

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
