## [Decision Letter]

Thank you for submitting your article "Parallel Visual Circuitry in a Basal Chordate" for consideration by *eLife*. Your article has been reviewed by Ronald Calabrese as the Senior Editor, a Reviewing Editor, Oliver Hobert, and two reviewers. The reviewers have opted to remain anonymous.

The reviewers have discussed the reviews with one another and the Reviewing Editor has drafted this decision to help you prepare a revised submission.

The reviewers – and the Reviewing Editor – agree that the manuscript reports an interesting, exciting set of findings that provide new insight into how visual systems evolve. However, there is also agreement that the evidence behind the GABA receptors being involved in the behavioral response to dimming is entirely indirect, and would be substantially strengthened by a pharmacological parallel to the Glutamate receptor antagonist data. That is, according to the disinhibition model, acute blockade of GABA(A) receptors with a pharmacological antagonist should produce a "hyperactive" movement phenotype akin to the *frm* mutant animal, but one that should still be capable of phototaxis (but not a dimming response). Such a result would provide an elegant "double dissociation" that would parallel the findings with the AMPA receptor antagonist.

There is also agreement that the manuscript requires an extensive revision to the Introduction that puts the work in a broader context. At present, the manuscript begins largely with a description of the *Ciona* connectome, in relation to other complete connectomes, and then plunges directly into a more detailed description of ganglia, cells and synapses. A broader audience could be engaged by the work if the authors identified the key question of interest, and provides some of the background material currently found in the Discussion section, before diving into the pertinent details.

*Reviewer #1:*

How the functional architecture of visual systems has evolved to subserve different behavioral goals is a fundamental question of broad interest. At present, while we have a deep understanding of visual system organization in a few experimental models, such a fundamental question can be enriched through the exploration of evolutionarily divergent organisms. In this context, Smith and colleagues integrate a new description of neurotransmitter expression patterns, ultrastructural connectivity, pharmacology and behavior to derive new insights into the architecture of the Ascidian *Ciona* visual system.

First, by mapping RNA expression patterns onto neurons spanning the *Ciona* nervous system using a combination of HCR in situs and image registration, they assign neurotransmitter types to many neurons. Importantly, these studies reveal three classes of ocellus photoreceptors – one that uses glutamate as a transmitter, one that uses GABA, and one that appears to release both. Next, using a glutamate receptor antagonist, they demonstrate that blockade of signaling from glutamatergic photoreceptors blocks phototaxis, but does not affect a second behavior evoked by transient dimming. Finally, consistent with the idea that a subset of photoreceptors could control the dimming response by depolarizing to darkness, and releasing GABA, the authors describe a mutant in which visual input to motor pathways is disrupted, leading to an animal that swims constitutively.

Overall, this manuscript reports an interesting, exciting set of findings that provide new insight into how visual systems evolve. I find the idea that there might be photoreceptors that appear to hyperpolarize to light and release GABA particularly exciting, and it will be fascinating to learn more about how these photoreceptors are related to retinal and pineal photoreceptors in vertebrates. However, I do feel that the evidence behind these receptors being involved in the behavioral response to dimming is entirely indirect, and would be substantially strengthened by a pharmacological parallel to the Glutamate receptor antagonist data. That is, according to the disinhibition model, acute blockade of GABA(A) receptors with a pharmacological antagonist should produce a "hyperactive" movement phenotype akin to the *frm* mutant animal, but one that should still be capable of phototaxis (but not a dimming response). Such a result would provide an elegant "double dissociation" that would parallel the findings with the AMPA receptor antagonist.

*Reviewer #2:*

The fact that there is a full map of connections in *Ciona* provides a great opportunity to dissect circuits. Even better, the tools are there to perform some genetic and pharmacological pertubations, and evaluate effects on behavior. This study begins to exploit these features in a study of the *Ciona* visual system. The authors dug deeper into two circuits that begin with photoreception. They used transgenic reporter animals and in situ hybridization to define the use of two classical neurotransmitters, glutamate and GABA. Surprisingly, one type of photoreceptor uses GABA, an inhibitory neurotransmitter not previously described as used by photoreceptors in any species. From the known connections, they also make a case for how the two circuits are connected, and further suggest that one of the circuits is disinhibitory, perhaps along with other sensory inputs, for oscillatory swimming behavior. Through the use of a specific antagonist for a glutamate receptor they are able to show that one of the photoreceptor circuits is involved with detection of the direction of light (phototaxis), using a behavioral assay. Interestingly, inhibition of phototaxis has no effect on the other circuit, which detects dimming. However, it is likely that there is cross talk between the two photoreceptor circuits, as suggested by the known anatomy.

Overall this study provides a very nice example of photoreceptor directed behavior as controlled by two different circuits. It provides food for though regarding the evolution of different types of visually guided behaviors and the use of different types of photoreceptors. Optogenetic manipulations and calcium imaging (tried by the authors but did not work due to technical limitations) would greatly add to this story, but as it stands it constitutes a very nice addition to our understanding of a sensory circuit and behavior.

---

## [Author Response]

The reviewers – and the reviewing editor – agree that the manuscript reports an interesting, exciting set of findings that provide new insight into how visual systems evolve. […] A broader audience could be engaged by the work if the authors identified the key question of interest, and provides some of the background material currently found in the Discussion section, before diving into the pertinent details.

In our revised manuscript we have thoroughly addressed the reviewers concerns and have included extensive new data from behavioral studies using a GABA receptor antagonist (Figure 7 in the revised manuscript, and related text). As you will read in the text, our results with the GABA receptor antagonist (picrotoxin) agree thoroughly with our disinhibition model (and with our observations of the *frm* mutant). The use of the GABA receptor antagonist was an excellent suggestion, and we feel that the results presented here greatly strengthen our model. As you will see in Figure 7, picrotoxin (like the *frm* mutant) leads to increased spontaneous swimming. Moreover, picrotoxin also leads to a dramatic reduction in the dimming response. We then show with use of picrotoxin combined with the AMPAR antagonist perampanel that the residual dimming response is due to parallel activation of the excitatory circuit. Finally, we show that picrotoxin-treated larvae are still capable of phototaxis. However, we observed that the phototaxis ability of the picrotoxin-treated larvae was somewhat dampened in comparison to controls, which we attribute to excitotoxicity of prolonged picrotoxin exposure (Movie5 documents the toxicity of prolonged picrotoxin exposure).

We have also extensively rewritten the Introduction along the lines suggested by the reviewer. Additionally, as requested, we have included in the text the number of animals tested using the pOpsin1/VGAT Kaede combination (n=5). Finally, we have collected additional data on neurotransmitter use by cells of the motor ganglion. These additional data are presented in revised versions of Figure 4 and Figure 4-figure supplement 1. Our conclusions regarding the minimal circuit are unchanged by this additional data; however, we are revising our neurotransmitter assignment to the anterior pair of ACINs. This reassignment was undertaken after consultation with Kerrianne Ryan (author of the *Ciona* connectome manuscript). We also include an approved personal communication from Dr. Ryan in this section.